# BigCodeArena: Unveiling More Reliable Human Preferences in Code Generation via Execution

## Abstract

Crowdsourced model evaluation platforms, such as Chatbot Arena, enable real-time evaluation from human perspectives to assess the quality of model responses. In the coding domain, manually examining the quality of LLM-generated content is extremely challenging, as it requires understanding long chunks of raw code and deliberatively simulating code execution. To this end, we introduce BigCodeArena, an open human evaluation platform for code generation back-ended with a comprehensive and on-the-fly execution environment. Built on top of Chatbot Arena, BigCodeArena features to enable the execution of LLM-generated code and allows humans to interact with the execution process and outcomes. We collected over 14K raw code-centric conversation sessions across 10 widely used LLMs, spanning 10 languages and 8 types of execution environments. Among these conversations, we identify more than 4.7K multi-turn samples with pairwise human preference. Further analysis uncovers the underexplored preferences of LLMs in fine-grained domains characterized by tasks, languages, and frameworks. To systematically examine code understanding and generation capabilities of frontier LLMs, we curate two benchmarks based on the collected data, namely BigCodeReward and AutoCodeArena. For BigCodeReward, we postprocess the 4.7K conversations and evaluate the consistency between reward models and human preference. The evaluation shows that most LLMs have superior performance in judging coding preferences when the execution results are given. Inspired by the findings, we propose AutoCodeArena, an automatic Elo rating benchmark designed to assess the coding quality of LLMs without humans. We find that proprietary LLMs like GPT-5, Claude-Sonnet-4, and Claude-Opus-4 still lead the performance in code generation among the recent emerging models. To democratize transparent evaluation of code generation in the wild, we aim to establish BigCodeArena as a long-term project.

## 1 Introduction

Large Language Models (LLMs) have demonstrated impressive capabilities across dialogue, reasoning, and code generation tasks (Zhao et al., 2023). As these systems rapidly evolve, robust evaluation has become essential. Human-in-the-loop platforms such as Chatbot Arena (Chiang et al., 2024) address this need by collecting pairwise human preferences on model responses, providing an off-the-shelf platform to assess open-ended outputs. Beyond text, recent efforts also engage humans in evaluating visual content from generative models (Jiang et al., 2024; Li et al., 2025b; Chou et al., 2025), offering new insights into multimodal models through real-world interactions.

Existing crowdsourced evaluators work well for common dialogue and visual content that are intuitive to compare. However, when evaluating long chunks of model-generated code, understanding the code semantics and reasoning about its runtime behaviours and non-functional properties are mentally exhausting and often demand specific expertise (Zhuo et al.; Jain et al.; Liu et al., 2024). Additionally, empirical studies confirm that humans often misjudge correctness without running the code (Détienne & Soloway, 1990; Lopez et al., 2008; Hassan et al., 2024). Exemplified by Figure 1, by reading the raw code, it is unclear which code snippet is superior; however, with execution feedback, it becomes visually obvious that model B is producing a higher-quality frontend.

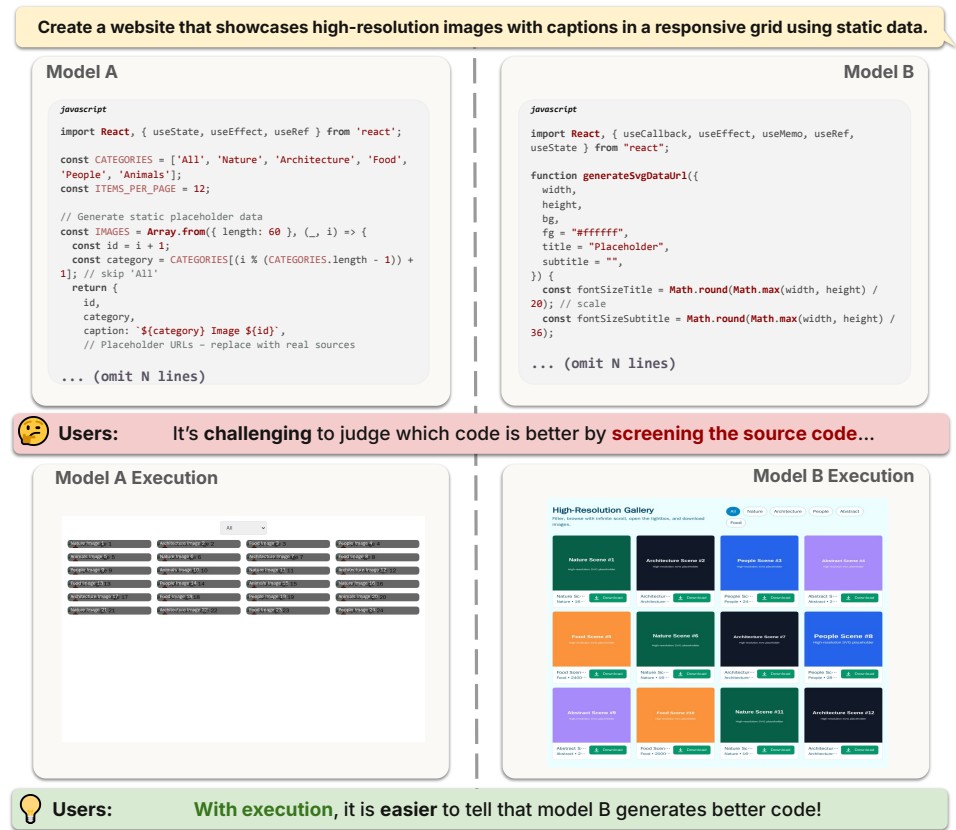

Figure 1: BIGCODEARENA enables users to evaluate code snippets based on execution outcomes.

***Here, we argue that the execution feedback is essential for humans to judge code quality reliably.*** Based on the aforementioned observation, we introduce BIGCODEARENA, a new open evaluation platform to collect human preferences of LLM-generated code, enabling on-the-fly compilation and execution of generated code, interactive debugging through code editing, and direct UI interaction. These features allow users to engage with program behavior rather than static snippets, providing a more robust and reliable evaluation of LLM outputs.

BIGCODEARENA has now been deployed for over five months, which enables us to collect more than 14K crowdsourced conversation sessions on code generation tasks spanning 10 languages (e.g., Python, Golang, JavaScript) and 8 execution environments (e.g., PyGame, React, Mermaid). This benchmark offers a glance at users' interaction and preference when using 10 frontier LLMs, such as o3-mini (OpenAI, 2025), Claude-3.5-Sonnet (Anthropic, 2025), and GPT-4o (Hurst et al., 2024). Analysis of the collected data highlights diverse usage scenarios, including Web Design, Game Development, Diagram Creation, Creative Coding, Scientific Computing, and Problem Solving, and reveals differences in language strengths and framework preferences across models. Together, these findings position BIGCODEARENA as an open, dependable, and execution-featured platform for advancing the evaluation of LLMs.

To facilitate future research on systematic evaluation of code generation, we further release two benchmarks: BIGCODEREWARD and AUTOCODEARENA. Like RewardBench (Lambert et al., 2025), BIGCODEREWARD measures how closely reward models (Ouyang et al., 2022) align with human judgments in code evaluation. On the other hand, AUTOCODEARENA aims to automate crowdsourced evaluation of BIGCODEARENA in the style of Arena-Hard-Auto (Li et al., 2025a). Evaluating more recent LLMs with these benchmarks yields three main findings. *First*, there is no obvious gap between proprietary and open LLMs when judging the code quality. *Second*, most LLM judges of code generation become substantially more reliable when execution results (e.g., UI screenshots) are available, reinforcing our motivation for building BIGCODEARENA. *Third*, we find that GPT-5 currently leads the quality of code generation among the frontier LLMs, while Claude-Sonnet-4 and Claude-Opus-4 are tied in second place.

Table 1: BIGCODEARENA is the first crowdsourced evaluation platform on code generation via execution, publicly releasing crowdsourced preference data. We collect 14K raw conversation sessions (analyzed in Appendix F) and a high-quality subset of 4.7K multi-turn conversations with human preference (discussed in Section 4). **Real-time**: whether the evaluation requires real-time interactions; **Multi-turn**: whether the evaluation supports multi-turn conversations; **Verified**: whether the output of LLMs is verified by human experts and executable environments; **Codebase**: whether the codebase is publicly available; **Data**: whether the data is fully open. We note that WebDev Arena only focuses on the Next.js framework for web design.

| Domain | Evaluation Platform | Real-time | Multi-turn | Verified | Codebase | Data | # Instances |
|--------|---------------------|-----------|------------|----------|----------|------|-------------|
| Vision | 3D Arena (Ebert, 2025) | ✗ | ✗ | ✗ | ✓ | ✗ | 0 |
| Vision | Genai Arena (Jiang et al., 2024) | ✓ | ✗ | ✗ | ✓ | ✗ | 0 |
| Vision | K-sort Arena (Li et al., 2025b) | ✓ | ✗ | ✗ | ✓ | ✗ | 0 |
| Vision | Vision Arena (Chou et al., 2025) | ✓ | ✗ | ✗ | ✓ | ✓ | 919 |
| Speech | S2S-Arena (Jiang et al., 2025) | ✗ | ✗ | ✗ | ✗ | ✗ | 0 |
| Text | Chatbot Arena (Chiang et al., 2024) | ✓ | ✓ | ✗ | ✓ | ✓ | 106K |
| Search | Search Arena (Miroyan et al., 2025) | ✓ | ✓ | ✗ | ✓ | ✓ | 24K |
| Code | WebDev Arena* (lma, 2025b) | ✓ | ✓ | ✓ | ✗ | ✓ | 10.5K |
| Code | Copilot Arena (Chi et al.) | ✓ | ✗ | ✗ | ✓ | ✓ | 114 |
| Code | BIGCODEARENA (Ours) | ✓ | ✓ | ✓ | ✓ | ✓ | 14K (4.7K) |

We summarize our contributions as follows:

- We present BIGCODEARENA, a new platform for human-in-the-loop evaluation of code generation, featuring real-time execution and interactive UI engagement.

- We host BIGCODEARENA over five months, yielding 14K crowdsourced conversation sessions on various code generation tasks and diverse usage of programming languages and frameworks.

- Among the collected conversations, we identify a subset of 4.7K multi-turn pairwise conversations and conduct a detailed investigation of user performance when using the models for code generation.

- On top of the collected conversations, we release two preference-focused benchmarks for code evaluation: BIGCODEREWARD, for assessing model alignment with human judgments, and AUTOCODEARENA, for automating output evaluation via LLM-as-a-Judge. Our extensive experiments demonstrate both the utility of these benchmarks and the advancements of recent LLMs.

## 2 BIGCODEARENA

As shown in Table 1, most existing crowdsourced evaluation platforms are closed-source (both in codebase and data), limiting transparency and verifiability. Moreover, few platforms focus specifically on code generation. To address these limitations, we present BIGCODEARENA, the first fully open-source human evaluation platform for LLM-generated code via execution. BIGCODEARENA extends Chatbot Arena by enabling code execution and user-driven testing, with more details in Appendix E.

### 2.1 MOTIVATION

LLM-generated code often appears syntactically correct while failing at runtime or misinterpreting the intended task. Traditional text-based crowdsourcing platforms such as Chatbot Arena (Chiang et al., 2024) present pairwise model responses for human voting, which works well for natural language tasks but fails to capture the complexities of code evaluation. Correctness frequently requires execution (Chen et al., 2021), since even minor changes can cause drastically different behaviors. Ensuring alignment with the prompt further requires a deep understanding of the task beyond surface fluency. Figure 1 illustrates this gap between static and interactive evaluation. For example, when prompted to build a responsive gallery website, both models generate code that looks reasonable in source form. However, it is difficult for users to determine which is better simply by reading the snippets. Once executed, though, it becomes clear that Model B produces a functional and visually appealing high-resolution grid, while Model A falls short. Rigorous evaluation thus requires execution feedback, not just static inspection, to capture the true quality and usefulness of code.

## 2.2 SYSTEM DESIGN

At a high level, BIGCODEARENA adopts a head-to-head evaluation setup. Given a user prompt, the platform presents two anonymized responses from different LLMs together with the execution results of extracted code snippets. These results are rendered either as interactive artifacts (e.g., applications, web pages) or as static outputs (e.g., text, images). Unlike Chatbot Arena, where judgments are made from a single static display, BIGCODEARENA grounds evaluation in observable behavior and functional outcomes. Users can test execution results, explore program behavior, and edit the extracted code to assess correctness and robustness.

The system itself consists of a lightweight web-based frontend and a secure, modular backend, built on Gradio[1] and E2B[2]. The frontend supports syntax-highlighted code display, editing, dependency configuration, and execution result rendering. The backend manages dependency resolution, installs required packages, executes code in isolated sandboxed environments, and returns execution results. In addition to side-by-side chat comparisons inherited from Chatbot Arena, BIGCODEARENA also supports a one-sided chat mode to enable testing of model-specific features. Together, these components enable evaluation that goes beyond appearance, allowing users to judge which model response not only looks correct but also runs and fulfills the intended task.

Similar to Chiang et al. (2024), BIGCODEARENA allows users to ask questions and receive answers from two anonymous LLMs. After reviewing both responses, users vote for their preferred answer, with model identities revealed only after voting. To collect balanced crowdsourced human evaluations across all models, we implement a weighted sampling strategy for model pair selection. Initial participant models receive equal weights, while new entrants are temporarily upweighted to gather sufficient comparative data. Formally, with $M$ models $\{1, \ldots, M\}$ and sampling weights $w_i$, the probability of selecting pair $(i, j)$ is

$$p(i, j) = \frac{w_i \cdot w_j}{\sum_{k < \ell} w_k \cdot w_\ell}. \tag{1}$$

Here, $w_i$ is uniform for established models and higher for new entrants. This approach ensures fair exposure across models and generates more stable preference signals over time. A key challenge in evaluation platforms is avoiding bias from artifacts such as response latency or execution speed. In code generation tasks, users might inadvertently prefer faster responses even when quality is lower. To mitigate this bias, BIGCODEARENA ensures that both model outputs are displayed simultaneously only after both models have completed generation and execution, so that user preferences reflect response quality rather than generation speed.

## 3 DEPLOYMENT

**Setup** BIGCODEARENA was advertised in open-source communities. Following prior setups (Chiang et al., 2024; Chi et al.; Chou et al., 2025), participants were not compensated but received free access to state-of-the-art models. Because collecting preference data in the wild is time-consuming, we also recruited 15 volunteer experts from the community (see Appendix A) with diverse programming expertise. To ensure annotation quality, we provided detailed guidelines and asked them to use varied prompts. Volunteers were required to conduct multi-turn pairwise conversations with at least two user–model exchanges. In addition to overall preference votes (four classes: Model A/B Better, Tie, Both Bad) suggested in Chiang et al. (2024), we included optional subcategories such as correctness, efficiency, explainability, maintainability, and UI/UX design. Although some aspects, like efficiency, are hard to quantify, we encouraged annotators to provide rationales for reproducibility. Alongside preference judgments, we logged user inputs, sandbox environments, execution results, and interaction activities, which, though sometimes noisy, give insight into user interests.

**Data Collection** We choose 10 frontier LLMs (detailed in Appendix H) at the time of deployment (February, 2025), covering both open and proprietary models specializing at coding: Llama's Llama-3.3-70B (Grattafiori et al., 2024), Alibaba's Qwen2.5 (Qwen2.5-72B-Instruct and Qwen2.5-Coder-32B-Instruct) (Yang et al., 2025; Hui et al., 2024), OpenAI's GPT-4o (Hurst et al., 2024), OpenAI's o

---

[1] https://gradio.app/
[2] https://E2B.dev/

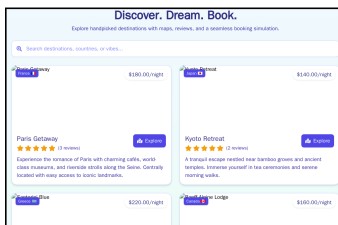

(a) **Web Design**: *React* for the travel plan.

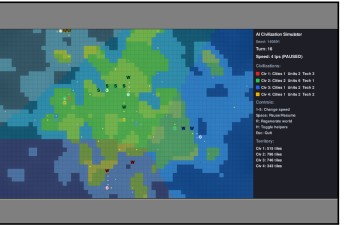

(b) **Game Development**: *PyGame* for AI civilization.

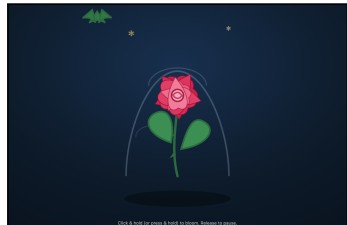

(c) **Creative Coding**: *Core Web* for a watery rose at night.

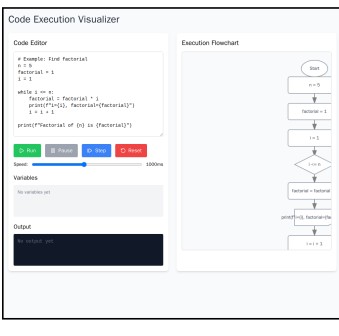

(d) **Diagram Creation**: *Vue* for deployment workflow visualization.

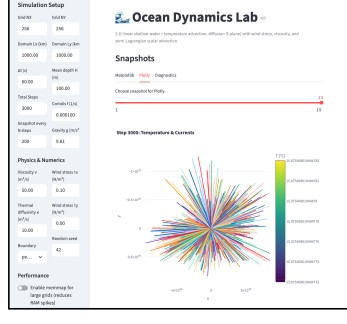

(e) **Scientific Computing**: *Streamlit* for Ocean simulation.

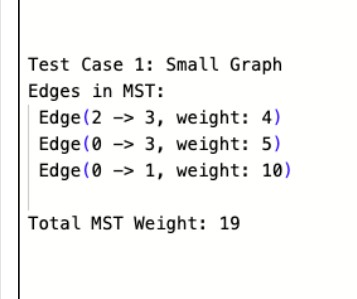

(f) **Problem Solving**: *Interpreter* for Kruskal's Algorithm for MST.

Figure 2: Examples of programming topics, prompts, and execution results in BIGCODEARENA.

series (o1, o1-mini, o3-mini) (Jaech et al., 2024), Anthropic's Claude-3.5-Sonnet (Anthropic, 2025), and Google's Gemini-2.0 (Gemini-2.0-Pro and Gemini-2.0-Flash) (Google Research & Google DeepMind, 2024). To ensure the diversity of collected data, we set the temperature to 0.7 and top-p to 0.95 by default, though these settings are adjustable by users. Over the course of 5 months, we collected over 14,123 conversations from more than 500 unique IP addresses.

**Languages and Environments** BIGCODEARENA currently supports 10 languages (Python, JavaScript, TypeScript, HTML, C, C++, Java, Go, Rust, and Markdown), and 8 execution environments (React, Vue, Core Web, Streamlit, PyGame, Gradio, Mermaid, and Interpreter). These are chosen to balance coverage of the most widely used languages in software development with frameworks that enable interactive and UI-oriented applications, which are particularly relevant for evaluating execution-based code generation. Python and interpreter-based workflows are emphasized given their ubiquity in data science and rapid prototyping, while the inclusion of web and game frameworks ensures diversity in coding tasks and real-world deployment scenarios. The descriptions of supported execution environments can be viewed in Appendix E.

**Topic Modeling** From the 14K collected conversation sessions, we attempted to automatically cluster user prompts following the pipeline in Chiang et al. (2024), but the results did not yield clear topic boundaries. To provide a more interpretable categorization, four of the authors manually inspected 50% of collected prompts (randomly sampled) and identified six recurring topics (with examples in Figure 2): (1) *Web Design*, focusing on building and styling websites; (2) *Game Development*, involving the creation of interactive games; (3) *Diagram Creation*, generating visual representations of systems or ideas; (4) *Creative Coding*, using code for artistic or experimental purposes; (5) *Scientific Computing*, applying code to numerical and data-driven tasks; and (6) *Problem Solving*, where logical reasoning and algorithms are central.

## 4 MODEL RANKING

Based on the data analysis of 14K conversations in Appendix F, we now examine the voting outcomes that define model rankings. To ensure data quality, we filter out pairwise conversations with fewer than two turns or without code execution, resulting in 4,731 voting samples, with each evaluated

model receiving at least 700 votes. Aggregating these into Elo ratings yields a leaderboard that reflects relative model strengths while accounting for uncertainty and context. This shifts our analysis from descriptive interaction patterns to quantitative comparisons of model performance. We provide detailed analysis such as programming topics and comparisons to previous evaluations in Appendix G.

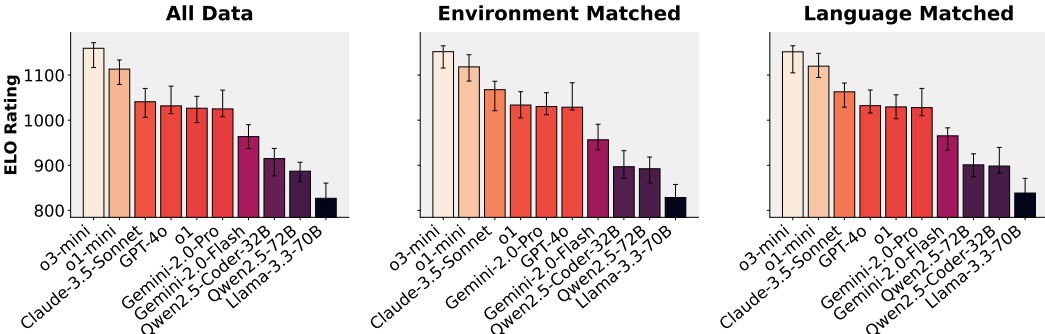

Figure 3: Elo ratings of models under three pair-sampling constraints: All Data (*left*), (Execution) Environment Matched (*middle*), and Language Matched (*right*). These settings progressively control runtime and language factors by using all data, matching environments, and restricting to the same language.

We construct a leaderboard from user preference judgments collected via pairwise comparisons. Let $n$ be the number of sessions and $M$ the number of models. For each session $i \in [n]$, the indicator $X_i \in \{-1, 0, 1\}^M$ encodes model positions ($\{-1, 1\}$ for Model A/B Better, and $\{0\}$ for Tie and Both Bad), while the outcome $Y_i \in \{0, 1, 0.5\}$ records wins, losses, or ties. Following prior work (Chiang et al., 2024; Chi et al.; Chou et al., 2025), we apply the Bradley-Terry model (Bradley & Terry, 1952) to estimate relative strengths $\boldsymbol{\beta} \in \mathbb{R}^M$. The Bradley-Terry model assumes that the probability $p_{ij}$ that model $i$ beats model $j$ can be modeled as:

$$p_{ij} = \frac{e^{\beta_i}}{e^{\beta_i} + e^{\beta_j}}. \tag{2}$$

To capture statistical uncertainty, we use 100 bootstrap resamples to construct 95% confidence intervals. Models are ranked by median bootstrap ratings, with intervals indicating significance.

Based on 4.7K multi-turn sessions involving 10 models (Figure 3), we analyze three evaluation settings: (1) All Data, (2) Environment Matched, and (3) Language Matched. These control runtime and linguistic variability. Rankings are consistent across settings, with o3-mini and o1-mini forming a clear top tier across environments and languages. Claude-3.5-Sonnet follows closely, especially under language matching. GPT-4o, o1, and Gemini-2.0-Pro/Flash form a competitive mid-tier, though GPT-4o weakens slightly under language matching. Qwen2.5 models and Llama-3.3-70B lag behind, highlighting the performance gap between leading proprietary and open models.

## 4.1 DETAILED ANALYSIS

**Languages** To better understand how model performance varies across languages, we analyze model *overall* win rates broken down by language-specific prompts (left of Figure 4). Across the board, top-tier models such as o3-mini and o1-mini achieve dominant win rates in widely used languages like Python, Java, and C++, which are commonly encountered in real-world applications and benchmarks. Other frontier models such as Gemini-2.0-Pro exhibit strong performance in lower-resource languages like Rust, achieving the highest win rate in that category. These results suggest that different models display distinct expertise, with frontier models excelling in different niches. In contrast, bottom-tier models such as the Qwen2.5 variants perform inconsistently, with weaknesses in Rust and Go.

**Environments** To separate model ability from implementation and runtime factors, we analyze win rates by execution environments (right of Figure 4). o3-mini shows consistently strong performance

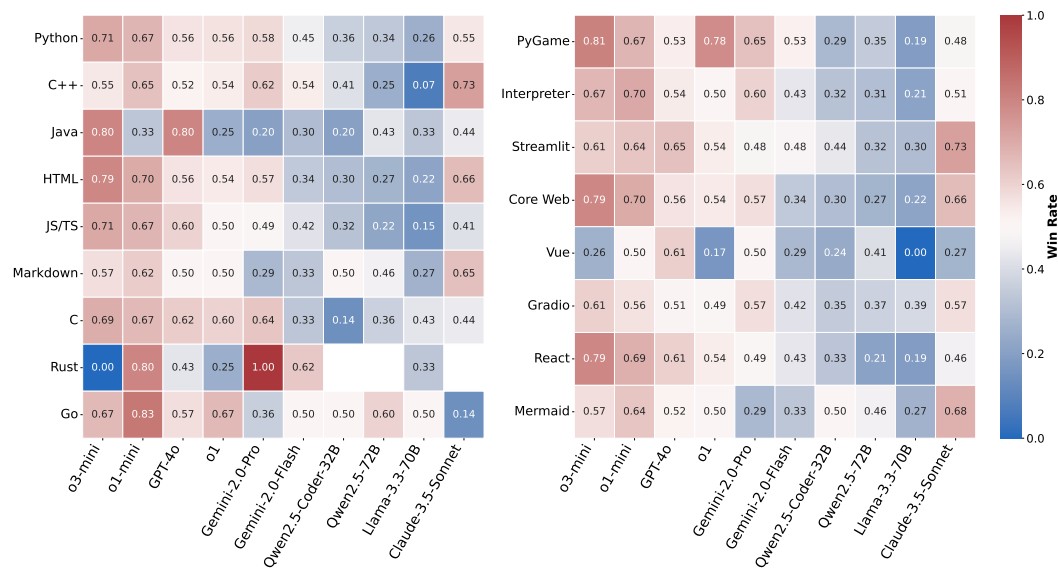

Figure 4: Overall win rate heatmaps (percentage of all pairwise comparisons won) of each model in the sessions across languages (*left*) and execution environments (*right*). For each category, we only keep models that appear in at least 3 conversation sessions.

across contexts such as React, Streamlit, Gradio, Core Web, and PyGame, indicating robustness to environmental variability. Claude-3.5-Sonnet and Gemini-2.0-Flash also show stable performance, though with lower win rates in more complex environments like Vue and Mermaid. In contrast, Qwen2.5 models, while competitive in some web frameworks, struggle in interactive and visualization-oriented execution such as PyGame, Vue, and Mermaid, which demand careful handling of control flow, graphics, and dependencies. These results suggest that despite high aggregate Elo scores, certain models remain brittle under realistic runtime constraints.

## 5 AUTOMATIC LLM EVALUATION WITH BIGCODEARENA

In this section, we introduce two benchmarks, BIGCODEREWARD and AUTOCODEARENA, for evaluating practical coding automatically. BIGCODEREWARD leverages the 4.7K human preference votes from Section 4 to study reward models across diverse coding tasks. For this benchmark, we postprocess the conversations by concatenating all user prompts with the final model response in each session, and use these aggregated instances as the evaluation inputs. AUTOCODEARENA, by contrast, provides automated comparisons in the style of BIGCODEARENA using 600 representative prompts, reducing reliance on long-term crowdsourced voting. More details of BIGCODEREWARD and AUTOCODEARENA are provided in Appendix I and Appendix J, respectively.

### 5.1 BIGCODEREWARD: EVALUATING REWARD MODELING FOR PRACTICAL CODING

**Motivation**  Reinforcement learning from human feedback (RLHF) sets a remarkable milestone in LLM training, where the models are trained to align with human preference (Bai et al., 2022). Instead of manually designing objective functions that capture nuanced human feedback, RLHF leverages preference data to train a reward model that serves as a proxy for human evaluation. While there have been a few works targeting the evaluation of reward models (Lambert et al., 2025; Malik et al., 2025), they mainly consider general domains. Execution-based code generation benchmarks like HumanEval (Chen et al., 2021) and BigCodeBench (Zhuo et al.) may be able to serve as a proxy for coding reward but still fail to capture the comprehensiveness of real-world scenarios. Therefore, we propose BIGCODEREWARD, the first benchmark for frontier code reward models.

**Setup**  We study how execution feedback affects reward models' ability to judge code quality, using accuracy as the metric. Unlike RewardBench (Lambert et al., 2025), models must choose

Table 2: Accuracy results (%) for reward models across task categories with/without execution outputs. "–" denotes without, "+" with execution. While *all* proprietary models benefit, *some* open LLMs drop in accuracy of multimodal coding scenarios, suggesting instability and insufficient robustness when incorporating multimodal feedback.. Best results are shown in **bold**.

| Models | Web | | Game | | Creative | | Diagram | | Scientific | | Problem | | Overall | |
|---|---|---|---|---|---|---|---|---|---|---|---|---|---|---|
| | – | + | – | + | – | + | – | + | – | + | – | + | – | + |
| *Proprietary Models* | | | | | | | | | | | | | | |
| Claude-Sonnet-4 (Anthropic, 2025) | 59.1 | 62.4 | 58.1 | 66.2 | 64.5 | 67.4 | 55.0 | 71.8 | 52.7 | 59.9 | 52.0 | 57.9 | 56.7 | 62.3 |
| Claude-3.7-Sonnet (Anthropic, 2025) | 57.3 | 63.1 | 55.5 | 61.8 | 65.5 | **72.4** | 52.3 | 71.1 | 50.7 | 59.9 | 45.3 | 57.8 | 53.9 | 62.2 |
| Claude-3.5-Sonnet (Anthropic, 2025) | 61.2 | 63.7 | 58.5 | 63.7 | **69.5** | 69.7 | 54.4 | 63.1 | 56.6 | 62.7 | **57.3** | **64.2** | **59.7** | 64.1 |
| GPT-4.1 (OpenAI, 2025) | 57.4 | 60.3 | 59.2 | 65.0 | 64.7 | 64.2 | 55.0 | 67.8 | 52.5 | 58.4 | 45.2 | 54.5 | 54.7 | 60.0 |
| GPT-4.1-mini (OpenAI, 2025) | 55.1 | 60.3 | 56.5 | 63.0 | 59.7 | 64.5 | 45.0 | 61.7 | 51.4 | 60.4 | 45.9 | 55.7 | 52.8 | 60.1 |
| GPT-4o (Hurst et al., 2024) | 57.7 | 65.0 | 57.3 | 65.4 | 67.1 | 72.1 | 55.7 | 69.8 | 53.3 | 63.0 | 43.8 | 57.5 | 54.6 | 63.8 |
| GPT-4o-mini (Hurst et al., 2024) | 59.3 | 65.1 | 59.2 | 63.4 | 63.7 | 68.4 | 53.7 | 68.5 | 55.4 | 63.4 | 56.5 | 63.1 | 58.3 | 64.5 |
| *Open Source Models* | | | | | | | | | | | | | | |
| Gemma-3-27B (Team et al., 2025b) | 59.0 | 61.6 | 59.6 | 62.7 | 64.2 | 62.1 | 53.0 | 69.1 | 54.6 | 57.8 | 56.8 | 60.0 | 58.2 | 61.1 |
| Qwen2.5-VL-72B-Instruct (Bai et al., 2025) | **61.6** | **65.8** | 58.8 | **68.8** | 67.1 | 71.6 | 56.4 | **76.5** | **57.4** | **63.7** | 52.2 | 63.1 | 58.7 | **66.2** |
| Qwen2.5-VL-32B-Instruct (Bai et al., 2025) | 56.9 | 60.2 | 56.9 | 63.4 | 61.3 | 67.6 | 52.3 | 64.4 | 53.0 | 63.3 | 54.5 | 60.4 | 56.0 | 61.9 |
| InternVL3-78B (Zhu et al., 2025) | 60.0 | 42.9 | **60.0** | 47.0 | 65.5 | 45.0 | 49.7 | 39.2 | 54.8 | 46.6 | 50.7 | 54.5 | 57.3 | 46.8 |
| InternVL3-38B (Zhu et al., 2025) | 56.5 | 43.3 | 59.2 | 46.0 | 63.4 | 44.3 | 52.3 | 37.8 | 51.7 | 50.8 | 52.9 | 57.8 | 55.9 | 48.0 |
| GLM-4.5V (Hong et al., 2025) | 54.5 | 56.6 | 55.4 | 55.7 | 61.1 | 58.7 | 49.3 | 57.7 | 51.3 | 55.1 | 47.9 | 50.9 | 53.0 | 55.2 |
| MiMo-VL-7B-RL (Team et al., 2025a) | 50.7 | 49.8 | 51.7 | 54.2 | 57.7 | 58.3 | **57.4** | 60.7 | 47.8 | 54.9 | 40.5 | 42.5 | 49.0 | 50.7 |
| Kimi-VL-A3B-Thinking (Team et al., 2025c) | 46.1 | 46.4 | 44.5 | 47.6 | 47.7 | 54.1 | 39.5 | 55.0 | 45.3 | 49.2 | 39.3 | 38.5 | 44.2 | 46.2 |

among three options: Response A/B Better, or Tie (combining Tie and Both Bad in Section 3). We evaluate two settings: (1) without execution results and (2) with execution results, which may include textual logs, screenshots of webpages, interactive applications, or plots. As explained in Section 2, these multimodal outputs can convey user preferences beyond text. Because multimodal classifier-based evaluators are limited (Ng & Jordan, 2001), we focus on a wide range of open and proprietary generative models (see Appendix H). All models are evaluated under the LLM-as-a-Judge setting (Zhuo, 2024; Li et al., 2025a) with greedy decoding.

**Result Analysis** Table 2 reports results across six programming topics and overall averages, where we observe execution results generally improve accuracy. Proprietary models reach the highest scores, though Qwen2.5-VL-72B Instruct remains competitive among open-source options. Gains are largest in Diagram Creation and Game Development tasks, while smaller in Problem Solving. Some models show instability, for example InternVL3-78B dropping from 57.3% to 46.8% with execution results.

## 5.2 AUTOCODEARENA: AUTOMATING THE JUDGEMENT OF CODE GENERATION

**Motivation** While BIGCODEARENA provides a reliable and human-grounded way to evaluate the coding capabilities of advanced LLMs, the process is extremely resource-consuming, as it requires large-scale crowdsourced preference votes collected over long periods of time. Given the rapid pace of LLM development, where new models are released on a weekly rather than yearly basis, there is a pressing need for a more efficient benchmark that can track progress without incurring prohibitive human annotation costs. Inspired by Li et al. (2025a), we develop AUTOCODEARENA, an automatic benchmark that leverages strong LLMs to approximate human preferences by comparing model outputs against a baseline system. Based on Li et al. (2025a), we use Bradley & Terry (1952) model to produce model's the final model scores. We aggregate all pairwise comparisons against the baseline model and apply bootstrapping to estimate confidence intervals for each model's win rate relative to the baseline. Models with higher win rates are generally more preferred by humans.

**Setup** To enable efficient evaluation, we design a prompt-selection pipeline. Prompts are first categorized into six programming topics (via GPT-4.1-mini, see Section 3), ranked within each topic, and sampled proportionally to match the distribution of the 4.7K multi-turn conversations (Section 4). In total, 600 representative prompts are chosen, reflecting real-world usage rather than enforcing artificial balance. Most prompts do not specify programming languages, allowing models to select their own. Building on Section 5.1, we execute code snippets and provide outputs to judge models (Claude-3.7-Sonnet). To overcome rate limits and latency of remote sandboxes, we implement a local Docker-based execution system supporting multiple languages and frameworks in parallel (Appendix J). All models (listed in Appendix H) are run with greedy decoding, except reasoning models, which use temperature 1.0 with medium reasoning effort.

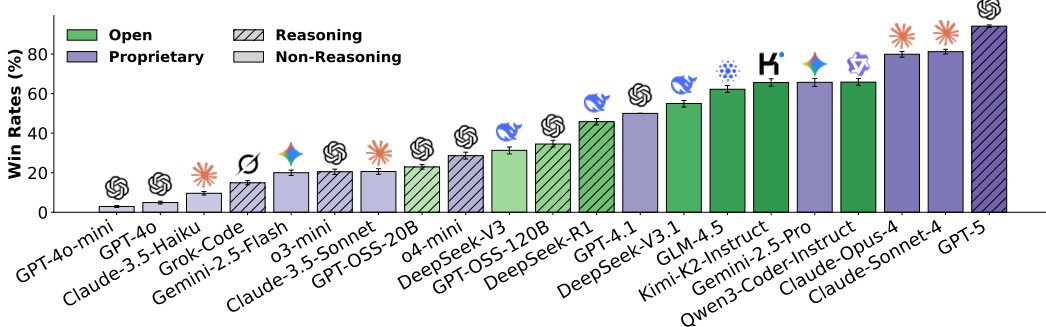

Figure 5: Overall performance of *more recent* LLMs on AUTOCODEARENA. We use GPT-4.1 as the baseline system and Claude-3.7-Sonnet as the judge. To avoid potential judgment bias toward self-generated responses, we exclude Claude-3.7-Sonnet from the rankings. GPT-4.1 is shown only to indicate the 50% win-rate baseline and is not compared against itself during evaluation.

**Result Analysis** The results in Figure 5 reveal several clear trends across the landscape of open and proprietary LLMs. Proprietary models continue to demonstrate a performance edge, with GPT-5 establishing a new state-of-the-art by a sizable margin. Both Claude-Opus-4 and Claude-Sonnet-4 also perform strongly, underscoring Claude's strength in reasoning-heavy tasks. Among open models, progress is visible but uneven. Open LLMs like Kimi-K2, GLM-4.5, and Qwen3-Coder form a leading cluster that significantly narrows the gap with mid-tier proprietary models. In contrast, models such as GPT-4.1 and Claude-3.5-Sonnet occupy the middle tier with moderate scores, while smaller models including GPT-4o-mini and Claude-3.5-Haiku lag substantially behind.

# 6 RELATED WORK

**Benchmarking Code Generation Quality** Most code generation benchmarks assess natural-language to code generation, where LLMs are prompted with natural language descriptions. Existing studies like HumanEval (Chen et al., 2021) and MBPP (Austin et al., 2021) evaluate coding capability through algorithm-specific problems that test whether generated code passes test cases. Practical benchmarks like DS-1000 (Lai et al., 2023) and BigCodeBench (Zhuo et al.) highlight library usage importance in code generation. Recent works emphasize multimodal code generation, such as generating webpages (Si et al., 2025; Yun et al., 2024) and plots (Wu et al., 2025). Different from existing benchmarks, we target dynamic evaluation for programming scenarios with execution results.

**Judging LLMs via Human Preference** Automatic benchmarks have been criticized due to limited scopes and potential contamination issues (Yang et al., 2023). As a result, human judgement is considered a natural and reliable metric to evaluate LLMs (Clark et al., 2021). To address limitations, Chatbot Arena (Chiang et al., 2024) computes model rankings by collecting human preference in pairwise comparisons. While there are code-specific platforms like Copilot Arena (Chi et al.), they fall short in application scope or do not open-source details. In this work, we aim to provide intuitive, transparent and reliable human evaluation of LLM-generated code via execution feedback.

# 7 CONCLUSION

We present BIGCODEARENA, an open evaluation platform for collecting human preferences on LLM-generated code via execution. Unlike prior platforms, it integrates real-time execution and interactive testing, enabling more reliable judgments of correctness, functionality, and intent alignment. Across 10 frontier LLMs and 4.7K crowdsourced conversations, we show that execution-based evaluation reveals issues overlooked by static comparisons. We further introduce two benchmarks: BIGCODERE-WARD, for measuring alignment with human preferences, and AUTOCODEARENA, for automating judgments with LLM-as-a-Judge. Our results highlight the value of execution signals, with GPT-5 leading overall and Claude models performing strongly. By constructing BIGCODEARENA and its benchmarks, we provide an open foundation for advancing robust, aligned code LLMs. We note that more related work and future work can be found in Appendix C and Appendix D, respectively.

## ETHICS AND REPRODUCIBILITY STATEMENT

Our work contributes to the societal benefit of responsible and transparent evaluation of code generation systems from a human-centered perspective. By releasing BIGCODEARENA and its associated benchmarks, we aim to equip researchers, practitioners, and policymakers with tools to better understand, compare, and improve large language models for software engineering. More reliable evaluations can accelerate the development of LLMs that are safer, more aligned with user intent, and ultimately more beneficial to communities that depend on trustworthy software systems. At the same time, we acknowledge and address the risks associated with this research. To enable evaluation, we collect user inputs and send them to various model API providers; while we make significant efforts to remove personally identifiable information (PII) during dataset preparation, complete elimination of sensitive content cannot be guaranteed. Additionally, although generated code is executed in a controlled, one-time remote sandbox environment (via the E2B cloud service), we cannot fully rule out the possibility of malicious or harmful code generation. We therefore emphasize both the limitations and the protective measures of our platform, highlighting the need for continued vigilance in mitigating cybersecurity and privacy concerns as the community builds on this work. Finally, we reflect on the compute and sustainability impact of our study. Unlike model training efforts that require substantial GPU clusters, BIGCODEARENA does not host LLMs locally, but instead leverages inference endpoints provided by multiple model API services. While this setup makes it difficult to precisely estimate $CO_2$ emissions, it reduces the direct energy footprint of our infrastructure. To support transparency and reproducibility, we document the specific inference endpoints used in Appendix H and will release all the artifacts (e.g., codebase, data, benchmarks, and experiment results) produced by this work. We encourage the broader community to continue investigating the trade-offs between large-scale evaluation, environmental sustainability, and accessibility.

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

# Appendix

C ONTENTS

## A  DATACARD

We follow (Bender & Friedman, 2018) to create the datacard for BIGCODEARENA, where we tend to summarize and centralize all information that might be relevant for the benchmark analysis.

**Curation Rationale**  This is detailed in Section 3.

**Language Variety**  Information about our annotators' nationalities will not be provided, as we do not have much knowledge of the public annotators. However, we confirm that all communications among the 15 volunteers recruited from the REDACTED community are in mainstream English (en-US). We note that the first language of these volunteers is not English, which can introduce some inaccurate expressions to the task prompts in the collected data.

**Curators Demographic**  The data annotation in BIGCODEARENA requires the great annotation effort of 15 Curators, who are involved in the process detailed in Section 3. They come from the following population:

- **Experience in Python Programming (Years)**:

  - 1-3: 7% (1/15)

  - 3-5: 60% (9/15)

  - 5+: 20% (3/15)

- **Experience in C Programming (Years)**:

  - 1-3: 27% (4/15)

  - 3-5: 27% (4/15)

  - 5+: 13% (2/15)

- **Experience in C++ Programming (Years)**:

  - 1-3: 27% (4/15)

  - 3-5: 27% (4/15)

  - 5+: 13% (2/15)

- **Experience in Java Programming (Years)**:

  - 1-3: 40% (6/15)

  - 5+: 20% (3/15)

- **Experience in Javascript/Typescript Programming (Years)**:

  - 1-3: 13% (2/15)

  - 3-5: 7% (1/15)

  - 5+: 7% (1/15)

- **Experience in Markdown (Years)**:

  - 1-3: 33% (5/15)

  - 3-5: 7% (1/15)

- **Experience in Rust Programming (Years)**:

  – 1-3: 27% (4/15)

- **Experience in Golang Programming (Years)**:

  – 1-3: 20% (3/15)

  – 3-5: 13% (2/15)

- **Experience in HTML Programming (Years)**:

  – 3-5: 20% (3/15)

- **Academic Background**:

  – Bachelor: 20% (3/15)

  – Master: 33% (5/15)

  – PhD: 47% (7/15)

**Text Characteristics**    This is detailed in Appendix F.

## B    DATA SHEET

Besides the provided Datacard, we follow the documentation frameworks provided by (Gebru et al., 2021).

### B.1    MOTIVATION

#### B.1.1    FOR WHAT PURPOSE WAS THE DATASET CREATED?

Our dataset aims to provide a thorough understanding of human preference on AI coding. Particularly, we focus on the challenges and practicability of the tasks, and pinpoint two main characteristics that few evaluations highlight: (1) Human understanding of the practical coding matter, and (2) Execution feedback is important to judge the code quality.This dataset will help stakeholders better understand the fundamental abilities and limitations associated with deploying LLMs.

### B.2    COMPOSITION/COLLECTION PROCESS/PREPROCESSING/CLEANING/LABELING AND USE

The answers are described in our paper as well as the GitHub repository: REDACTED.

### B.3    DISTRIBUTION

#### B.3.1    WILL THE DATASET BE DISTRIBUTED TO THIRD PARTIES OUTSIDE OF THE ENTITY (E.G., COMPANY, INSTITUTION, ORGANIZATION) ON BEHALF OF WHICH THE DATASET WAS CREATED?

No. Our dataset will be managed and maintained by the REDACTED community (REDACTED).

#### B.3.2    HOW WILL THE DATASET BE DISTRIBUTED (E.G., TARBALL ON WEBSITE, API, GITHUB)?

The evaluation dataset will be released to the public, and hosted on Hugging Face.

#### B.3.3    WHEN WILL THE DATASET BE DISTRIBUTED?

The dataset will be released after ICLR 2026.

### B.3.4 WILL THE DATASET BE DISTRIBUTED UNDER A COPYRIGHT OR OTHER INTELLECTUAL PROPERTY (IP) LICENSE, AND/OR UNDER APPLICABLE TERMS OF USE (TOU)?

Our dataset will be distributed under the Apache-2.0 license.

## B.4 MAINTENANCE

### B.4.1 HOW CAN THE OWNER/CURATOR/MANAGER OF THE DATASET BE CONTACTED (E.G., EMAIL ADDRESS)?

Please contact REDACTED (`REDACTED`) and the REDACTED Project (`REDACTED`), who are responsible for maintenance.

### B.4.2 WILL THE DATASET BE UPDATED (E.G., TO CORRECT LABELING ERRORS, ADD NEW INSTANCES, DELETE INSTANCES)?

Yes. If we include more tasks or find any errors, we will correct the dataset hosted on Hugging Face.

### B.4.3 IF OTHERS WANT TO EXTEND/AUGMENT/BUILD ON/CONTRIBUTE TO THE DATASET, IS THERE A MECHANISM FOR THEM TO DO SO?

For dataset contributions and evaluation modifications, the most efficient way to reach us is via GitHub pull requests. For more questions, contact REDACTED (`REDACTED`) and the REDACTED Project (`REDACTED`), who are responsible for maintenance.

## C  RELATED WORK

**Training LLMs on Code**   LLMs have significantly advanced the landscape of software engineering, including code completion (Chen et al., 2021) and program repair (Jin et al., 2023). Such models have been trained on a large-scale corpus of source code and able to capture of the code semantics. A series of code-specific LLMs like CodeGen (Nijkamp et al., 2023), StarCoder (Li et al.; Lozhkov et al., 2024), and InCoder (Fried et al., 2023), have been proposed to automate code generation in software development. However, these models have limited capabilities in understanding natural language and hence fail to follow complex instructions from humans (Zhou et al., 2023). Later, GPT-3.5 (Ouyang et al., 2022) was developed to specialize at both text and code generation, achieving superior capabilities in aligning human preference. Inspired by GPT-3.5, many LLMs have begun to merge code and text during training, such as Claude (Anthropic, 2025), Gemini (Team et al., 2023), Qwen (Bai et al., 2023), and DeepSeek (Bi et al., 2024). With the new training paradigm, LLMs demonstrate stronger cross-domain reasoning, improved instruction-following, and enhanced ability to ground code generation in natural language descriptions. This unified training has narrowed the gap between general-purpose LLMs and code-specialized models, enabling more reliable support for real-world programming tasks (Hou et al., 2024).

**Benchmarking Code Generation Quality**   Most of the code generation benchmarks assess the quality of natural-language to code generation, where LLMs are prompted with a natural language description or docstring. Existing studies like HumanEval (Chen et al., 2021) and MBPP (Austin et al., 2021) consider algorithm-specific code generation problems a good way to evaluate the coding capability, where the benchmarks test whether the generated code can pass a series of test cases. To make the evaluation more practical, researchers have proposed DS-1000 (Lai et al., 2023), APIBench (Patil et al., 2024), BigCodeBench (Zhuo et al.), which highlight the importance of library usage in code generation. To build beyond textual code, more recent works emphasize on the multimodal code generation, such as generating webpages (Si et al., 2025; Yun et al., 2024; Zhang et al., 2025), plots (Wu et al., 2025) and SVG (Rodriguez et al., 2025). Different from existing benchmarks, we target a more dynamic and interactive evaluation for any programming scenarios that can be presented with execution results.

**Judging LLMs via Human Preference**   Automatic benchmarks have been criticized due to the limited scopes and potential contamination issues (Yang et al., 2023). As a result, human judgement is

considered a more natural and reliable metrics to evaluate LLMs (Clark et al., 2021). Traditionally, humans may be asked to score the generation quality of LLMs based on some predefined rubrics (Freitag et al., 2021). However, conducting this kind of human studies is unscalable. Furthermore, the results can be varied when the evaluation criteria changes. To address the limitations, Chatbot Arena (Chiang et al., 2024) was created to compute the model rankings by collecting human preference in pairwise comparisons. It has attracted the community attention and turns into a long-term evaluation platform to keep evaluating new LLMs. Meanwhile, there are many other evaluation platforms like Vision Arena (Chou et al., 2025) for visual input understanding, Text-to-Image Arena (lma, 2025a) for image generation, and Search Arena (Miroyan et al., 2025) for LLM-based search. While Chatbot Arena has code-specific evaluation platforms like Copilot Arena (Chi et al.) and WebDev Arena (lma, 2025b), they fall short in the application scopes or do not open-source any details. In this work, we aim to provide a more intuitive, transparent and reliable human evaluation of LLM-generated code via execution feedback.

## D FUTURE WORK

BIGCODEARENA opens several promising directions for future research. First, although our platform is open-source and designed to be scalable across diverse execution environments, it currently supports only a limited set of languages and frameworks. We hope the community will contribute to expanding this diversity. Second, developing live versions of BIGCODEREWARD and AUTOCODEARENA would allow evaluation prompts to be continuously refreshed, drawing from both user inputs and LLM-generated tasks. Third, improving the reliability of evaluation is crucial: our current benchmarks rely on LLM-as-a-Judge using only initial screenshots, whereas future work could leverage LLM agents that actively interact with web applications for deeper assessment. Fourth, recording and utilizing user interaction trajectories in BIGCODEARENA may enable training LLMs to autonomously test and evaluate web applications in human-like ways. Fifth, advancing reward models for code generation remains an open challenge, as current systems still fall short of human-level perception and reasoning; better reward models will, in turn, support the development of more capable and aligned code LLMs. Finally, we envision BIGCODEARENA evolving into a comprehensive ecosystem that not only evaluates existing models but also serves as a training ground for next-generation code LLMs through continuous human-AI collaboration and real-world task discovery.

# E BIGCODEARENA

## E.1 SCREENSHOT

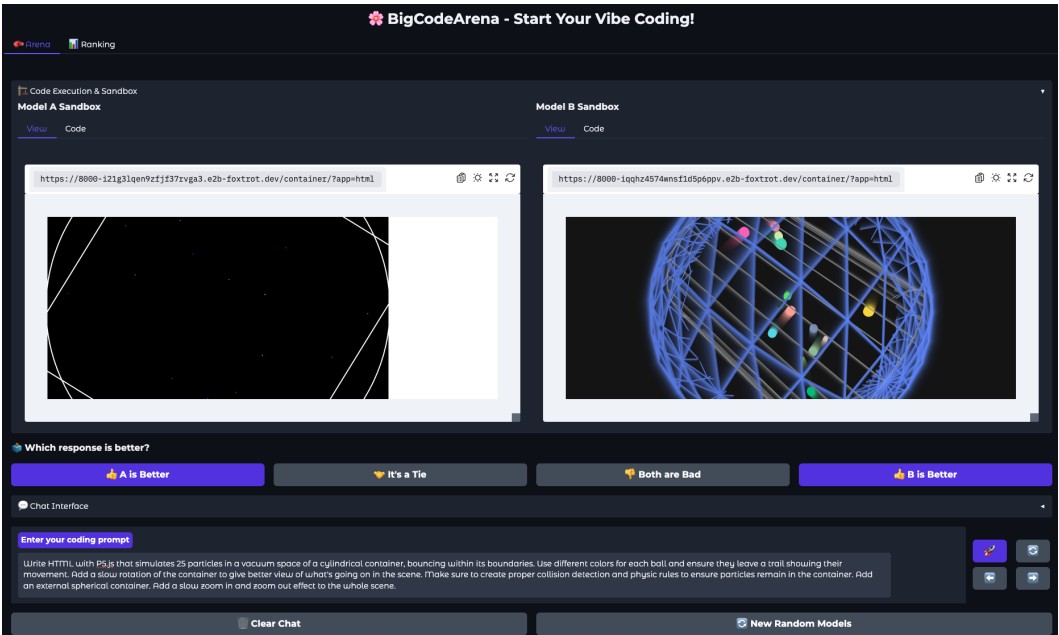

Figure 6: User interface of BIGCODEARENA. The left and right panes display outputs from two different models (A and B) in a code execution sandbox, while the bottom section allows users to view the prompt, inspect code, and cast comparative judgments on which model performed better. The example prompt is "Write HTML with P5.js that simulates 25 particles in a vacuum space of a cylindrical container, bouncing within its boundaries. Use different colors for each ball and ensure they leave a trail showing their movement. Add a slow rotation of the container to give better view of what's going on in the scene. Make sure to create proper collision detection and physic rules to ensure particles remain in the container. Add an external spherical container. Add a slow zoom in and zoom out effect to the whole scene."

## E.2 SYSTEM DESIGN

**User Interface** BIGCODEARENA provides an interface for direct pairwise comparison of anonymized model outputs in code generation tasks. By rendering responses in identical formats, the interface ensures that judgments are based on quality rather than model identity. The interface consists of three components: a unified input panel for coding prompts, dual response panels displaying outputs, and an integrated execution environment. Each response panel supports syntax highlighting, collapsible code blocks, and execution controls, allowing users to run code directly in the browser. This design shifts evaluation from subjective inspection to functionality-driven assessment. After reviewing and optionally executing outputs, users can vote for their preferred response, record a tie, or abstain if neither response meets quality standards. To further align with developer workflows, the interface supports in-place code editing for testing modifications and provides a conversation history for multi-turn interactions. These features capture real-world coding assistance scenarios where requirements evolve iteratively.

**Pipeline Modules** As displayed in Figure 7, the *code extraction* module analyzes each snippet to determine the runtime environment based on language identifiers in markdown code blocks. BIG-CODEARENA currently supports 10 languages (Python, JavaScript, TypeScript, HTML, Markdown, C, C++, Java, Golang, and Rust) and 8 frameworks (Core Web, React, Vue, Gradio, Streamlit, PyGame, Mermaid, and Interpreter); detailed descriptions are provided in Section E.4. The *sandbox execution* module parses imported packages, installs third-party dependencies, and executes code

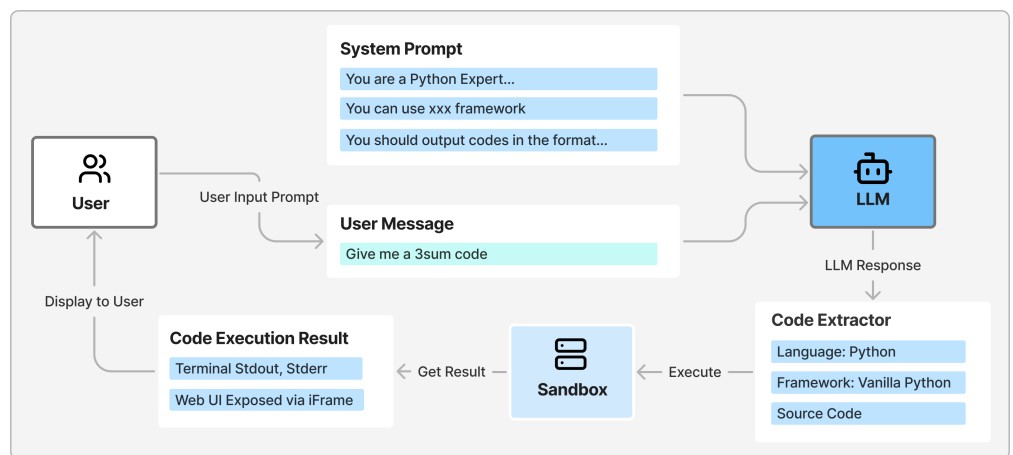

Figure 7: Overview of BIGCODEARENA Pipeline.

in containerized sandboxes. The system can create and terminate multiple isolated environments in parallel without affecting the platform. Execution is subject to time and memory limits to prevent infinite loops or resource exhaustion, reflecting real-world workflows where dependency management is critical. Finally, the *result display* module presents structured outputs, including logs, error traces, and runtime results, side by side for users. This design encourages testing of edge cases and verification of functionality before casting a vote. By embedding execution directly into the evaluation loop, BIGCODEARENA enables judgments based not only on presentation but also on correctness and practical utility.

### E.3 MODEL SAMPLING STRATEGY

A challenge in evaluation platforms is ensuring that user preferences are not biased by system-level artifacts such as streaming latency or execution time. In code generation settings, users may naturally prefer the model whose response appears faster or whose program executes earlier, even if this does not reflect true quality. To eliminate this confounder, BIGCODEARENA enforces strict synchronization: model responses are only displayed once both models have completed code generation and their outputs have finished execution in the sandbox environment. This guarantees that preferences are based solely on the quality and behavior of the generated programs.

For model sampling, we assign weights to balance coverage across models. By default, each model is assigned an equal weight, ensuring fair participation in comparisons. However, when new models are introduced to the arena, we temporarily upweight them to accelerate the collection of preference data. This strategy ensures that late-entering models accumulate a comparable number of votes, stabilizing the human preference and reducing variance in preference estimates.

Formally, given a set of $M$ models $\{1, \ldots, M\}$, each model $i$ is associated with a sampling weight $w_i$. The probability of selecting a model pair $(i, j)$ is then:

$$p(i, j) = \frac{w_i \cdot w_j}{\sum_{k < \ell} w_k \cdot w_\ell}, \tag{3}$$

where $w_i$ is uniform for established models and upweighted for late entrants until sufficient preference data is collected. By decoupling human judgments from latency artifacts and rebalancing model exposure, this approach provides fairer and more stable human preference signals for execution-enabled code generation.

### E.4 SUPPORTED EXECUTION ENVIRONMENTS

**Core Web**   The Core Web environment supports direct HTML execution with embedded CSS and JavaScript. Code is processed to replace placeholder URLs with SVG data URLs for self-contained rendering. The sandbox creates an HTML application directory, writes the provided code as an index.html file, and serves it through the E2B infrastructure's nginx proxy. This environment enables rapid prototyping of web interfaces without build processes or framework dependencies.

**React**   The React environment provides a complete React development stack with TypeScript support and Vite build system. The template includes React 18.3.1, React DOM 18.3.18, and TypeScript 5.6.2. User code replaces the default App.tsx component in a pre-configured React project template with Tailwind CSS 3.4.17 and PostCSS 8.5.1. The pipeline uses npm for dependency management with specific flags including `--prefer-offline`, `--no-audit`, `--no-fund`, and `--legacy-peer-deps` for robust installation. The build process executes `npm run build` with TypeScript compilation and serves the compiled application through the sandbox's web proxy.

**Vue**   The Vue environment mirrors the React setup but targets Vue.js 3.5.13 Single File Components with Vue Router 4.5.0. The template includes TypeScript 5.6.3, Vite 6.0.5, and Tailwind CSS 3.4.17. User code replaces the App.vue file in a pre-configured Vue project with Vite tooling. The system handles Vue-specific build processes and dependency management through npm, ensuring proper compilation of templates, scripts, and styles.

**Mermaid**   The Mermaid environment converts diagram syntax into interactive HTML visualizations. User-provided Mermaid code is embedded within a minimal HTML document that includes the Mermaid JavaScript library version 10.6.1 from CDN. The system supports configurable themes and security levels, with diagrams rendered client-side through the Mermaid initialization system.

**Gradio**   The Gradio environment enables rapid creation of machine learning interfaces and interactive demos. The template pre-installs Gradio through `uv pip install --system --upgrade gradio`. User code defines Gradio applications which are automatically configured to run on allocated ports with proper server settings. The pipeline uses uv for Python dependency management, installing packages with `--system` flag for global availability.

**Streamlit**   The Streamlit environment supports data science applications and interactive dashboards. The template pre-installs Streamlit through `uv pip install --system --upgrade streamlit`. User code is written as a Streamlit application script, with the pipeline using uv for dependency installation. Applications run in headless mode on port 8501 with `--server.headless true` and `--server.runOnSave false` flags.

**Interpreter**   The Interpreter environment executes code across multiple programming languages through the E2B code interpreter sandbox. The template includes build tools for C/C++ (`build-essential`), Java (`default-jdk`), Go (`golang`), and Rust (`rustc`). Python dependencies are managed through uv with `--system` installation, while npm handles JavaScript dependencies. The template pre-installs 101 top PyPI packages including pandas, matplotlib, scipy, numpy 1.26, and scientific computing libraries. Python code benefits from enhanced visual output capture through instrumentation of matplotlib and other visualization libraries, while compiled languages follow standard compilation workflows with `gcc`, `g++`, `javac`, `rustc`, and `go run` commands.

### E.5 SYSTEM PROMPT DESIGN

```
You are an expert Software Engineer, UI/UX designer, and product ←
    manager. Your task is to generate self-contained, executable code ←
    for a single file or block that can run directly in a sandbox ←
    environment. Feel free to ask questions or explain your reasoning.
If you do a great job based on the instructions, you will be rewarded ←
    with a high salary and a promotion.
```

```
Your code must be written using one of these supported development
    frameworks and environments:
- React (JavaScript/TypeScript)
- Vue (JavaScript/TypeScript)
- HTML (Vanilla HTML)
- Gradio (Python)
- Streamlit (Python)
- PyGame (Python)
- Mermaid (Markdown)
- Python Runner
- JavaScript Runner
- Command Line Code Runner (C/C++/Go/Java/Rust)

All web framework code (React, Vue, HTML) must be directly rendered in
    a browser and immediately executable without additional setup. DO
    NOT create separate CSS files
Python-based frameworks should be directly executable in a browser
    environment.
The code to be executed in Runners must be plain Python or JavaScript
    programs that do not require web UI frameworks or standard user
    input.

The code must be in the markdown format:
```<language>

```

Before you begin writing any code, you must follow these fundamental
    rules:
- You are NOT allowed to start directly with a code block. Before
    writing code, ALWAYS think carefully step-by-step
- Your response must contain a clear explanation of the solution you
    are providing
- ALWAYS generate complete, self-contained code in a single file
- You CAN NOT split your program into multiple files or multiple code
    blocks
- If you use any external libraries, make sure to specify them for the
    installation command in either `pip install` or `npm install`
- You prefer JavaScript over HTML
- Each code block must be completely independent. If modifications are
    needed, the entire code block must be rewritten
- When fetching data, you MUST use external libraries and packages,
    and avoid using placeholder URLs or URLs that require API keys
- Make sure the program is functional by creating a state when needed
    and having no required props
- Make sure to include all necessary code in one file
- There are no additional files in the local file system, unless you
    create them inside the same program
- Do not touch project dependencies files like package.json, package-
    lock.json, requirements.txt, etc

When developing with React or Vue components, follow these specific
    requirements:
- Use TypeScript or JavaScript as the language
- DO NOT use gray text color on a white background
- Make sure it can run by itself by using a default export at the end
    of the file
- DO NOT CALL `ReactDOM.render()` AT THE END OF THE FILE
- Use Tailwind classes for styling. DO NOT USE ARBITRARY VALUES (e.g.
    'h-[600px]'). Make sure to use a consistent color palette
- If you use any imports from React like `useState` or `useEffect`,
    make sure to import them directly
- Use Tailwind margin and padding classes to style the components and
    ensure proper spacing
```

```
- Various npm packages are available to be imported, e.g. `import {
    LineChart, XAxis, ... } from "recharts"` & `<LineChart ...><XAxis
    dataKey="name"> ...`
- Images from the web are not allowed, but you can use placeholder
    images by specifying the width and height like so `<img src="/api/
    placeholder/400/320" alt="placeholder" />`

For Python development, you must follow these constraints:
- For any programs that require user inputs, you MUST USE `gradio` or
    `streamlit`
- Choose suitable PyPI packages to be imported, e.g., `import pandas`
- Avoid using libraries that require desktop GUI interfaces, with the
    exceptions of `pygame`, `gradio`, and `streamlit` which are
    explicitly supported
- For PyGame applications, you have to write the main function as an
    async function like:
```python
import asyncio
import pygame

async def main():
    global game_state
    while game_state:
        game_state(pygame.event.get())
        pygame.display.update()
        await asyncio.sleep(0) # it must be called on every frame

if __name__ == "__main__":
    asyncio.run(main())
```

For HTML development, ensure that:
- All HTML code must be self-contained in a single file
- Include any necessary CSS and JavaScript within the HTML file
- Ensure the code is directly executable in a browser environment
- Images from the web are not allowed, but you can use placeholder
    images by specifying the width and height like so `<img src="/api/
    placeholder/400/320" alt="placeholder" />`

For Mermaid development:
- Write Mermaid diagrams directly using ```mermaid code blocks, e.g.:
```mermaid
graph TD;
    A-->B;
```

For Command Line Code Runner (C/C++/Go/Java/Rust), ensure that:
- ALWAYS generate complete, self-contained code in a single file.
    Avoid non-standard libraries.
- Your code should be able to be compiled and run directly.
- Your code must complete the task without any user inputs. It should
    not be long running.
- You should provide example test cases in the code and output the
    result to stdout or stderr.

The code must be in the markdown format:
```<language>

```
```

## E.6 SANDBOX INFRASTRUCTURE FOR BIGCODEARENA

The sandbox infrastructure in BIGCODEARENA is built upon the E2B platform, which provides a managed execution environment for code evaluation. The system operates through a centralized sandbox manager that handles the creation, lifecycle management, and resource allocation for various programming environments. Each sandbox instance is created with a specific template configuration that includes pre-installed language runtimes, build tools, and development dependencies. The core execution pipeline follows a unified approach where user code is injected into pre-configured project templates. For web frameworks like React and Vue, the system maintains dedicated project structures with TypeScript configurations, build tools, and styling frameworks. The sandbox manager ensures that each execution environment has the necessary dependencies installed and that the build processes complete successfully before serving the applications. Dependency management is handled through multiple package managers: uv for Python packages with `--system` installation flags, and npm for JavaScript dependencies with conflict-tolerant flags like `--legacy-peer-deps` and `--prefer-offline`. The infrastructure pre-installs a comprehensive set of 101 top PyPI packages including scientific computing libraries, web frameworks, and development tools to minimize cold-start latency. For interactive applications, the sandbox infrastructure provides background process management with timeout handling and error monitoring. Web servers are launched with specific port allocations and headless configurations, while the system monitors for startup errors and provides access through the E2B nginx proxy. The infrastructure also supports multiple programming languages through the E2B code interpreter sandbox, which handles compilation and execution for C, C++, Java, Go, and Rust code. The sandbox infrastructure emphasizes reproducibility and isolation by using fresh instances for each execution and maintaining consistent environment configurations across runs. This design ensures that code evaluation results are deterministic and that different submissions can be compared fairly within the benchmarking framework.

## E.7 CASE STUDIES ON UI INTERACTIONS

**Turn 1: Please build my new App Idea: A Virtual Event Planner and RSVP System**

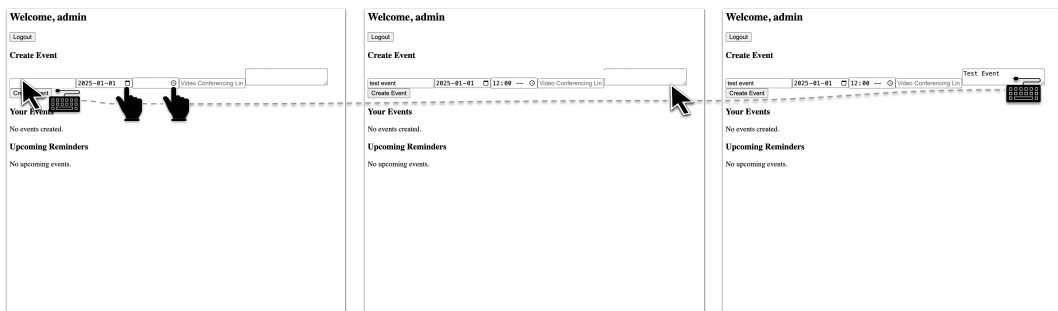

**Turn 2: The UI is very boring looking, can you please add a lot more emojis and fun graphics to engage the user. Please also include fun and exciting colors to make the user feel connected.**

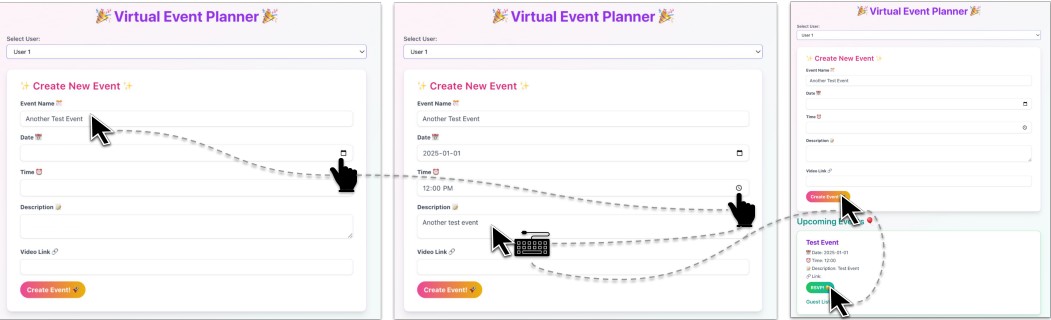

Figure 8: User interaction trajectories on initial functional testing and subsequent UI enhancement evaluation.

Figure 8 provides a case study on how the user interacts with the rendered webpage generated from the LLM-produced code snippet. The case study illustrates two distinct turns in the interaction trajectory, each of which reveals different testing intentions by the user.

In the first turn, the user engages with the minimal event creation form. The sequence of actions shows that the user selects a date from the calendar picker, specifies a time and a video conferencing link, and enters a simple placeholder event name. This trajectory indicates that the user is primarily interested in verifying the core functionality of the system. The focus lies on checking whether the form fields accept inputs correctly, whether the basic flow of creating an event works as expected, and whether the system registers the submitted data as a valid "test event." At this stage, the concern is functionality rather than aesthetics, and the test represents a validation of the underlying event creation logic.

In the second turn, the user provides explicit feedback that the user interface is "boring" and requests improvements in visual appeal, including emojis, engaging graphics, and colorful styling. The user's interactions shift toward evaluating the usability and design of the interface. The actions involve inputting richer event details such as a more descriptive event name, a textual description, and other contextual information. The user then triggers the creation of the event and verifies that it appears correctly under the list of upcoming events. This trajectory demonstrates that the user is testing not only whether events can be created and displayed but also how the interface supports user engagement and perceived connectedness. Through this progression, the case study highlights a natural transition from functional validation to user experience evaluation.

## F DATA ANALYSIS

In this section, we provide the analysis of the collected 14K conversations from two perspectives: (1) Conversation Characteristics, and (2) User Activities.

## F.1 CONVERSATION CHARACTERISTICS ANALYSIS

**Conversation Context** We first analyze the interaction statistics across roles. On average, user messages have 291.64 characters per turn. Model responses, by contrast, are substantially longer, consistent with their role in elaborating on queries and providing detailed explanations. In terms of language diversity, users employ a limited set of 9 natural languages, whereas the assistant generates outputs across a considerably broader range, covering 10 languages supported in our sandbox environments. This divergence underscores the model's multilingual capacity relative to the more localized communication behavior

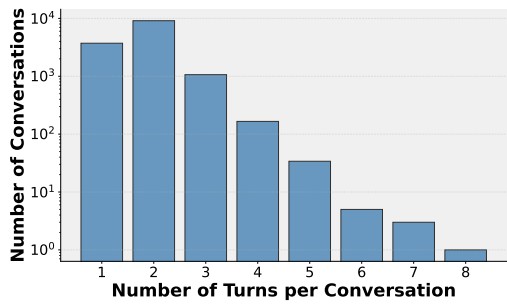

Figure 9: Distribution of conversation turns.

of users. The proportion of duplicate content is low for both roles, suggesting that interactions are varied and not dominated by repeated prompts or template-like responses. Overall, these findings indicate an asymmetry in conversation structure: user contributions are short and narrowly distributed across languages, while assistant outputs are longer, more diverse, and linguistically expansive.

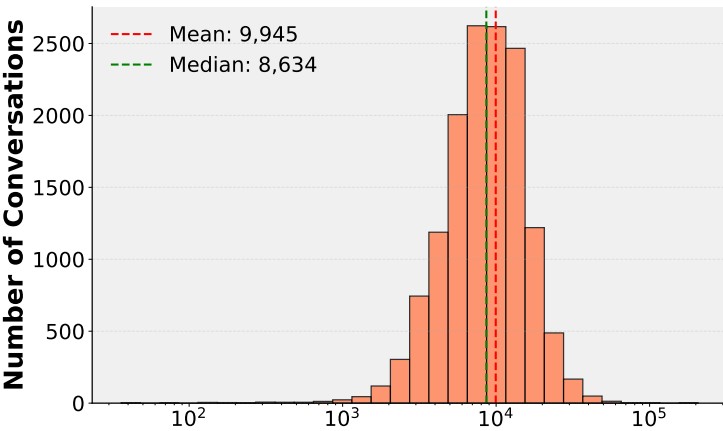

Figure 10: The distribution of character counts for conversations.

Figure 10 presents the distribution of conversation lengths measured in total characters across the BIGCODEARENA dataset. The histogram, displayed on a logarithmic scale to accommodate the wide range of values, reveals a log-normal-like distribution characteristic of natural language interactions. The majority of conversations cluster between 3,000 and 20,000 characters, representing typical coding assistance scenarios. This long-tail phenomenon is typical in code generation tasks where complex problems require extensive multi-turn interactions and detailed code outputs.

We next examine conversation length, summarized in Figure 9. The majority of conversations (76.1%) consist of exactly two turns, corresponding to a single user request followed by one model response. A smaller proportion (10.5%) are single-turn interactions. The mean length of conversations is 4.12 messages (2.06 turns), and 87.2% conclude within two to three turns. These results indicate that the predominant mode of interaction is short and goal-oriented, with users seeking targeted responses rather than engaging in extended dialogues. Longer conversations are comparatively rare, suggesting that while multi-step reasoning and iterative development are supported by the system, they do not represent the primary usage pattern. The distribution further implies that efficiency and directness are valued in typical use cases, with users preferring to resolve tasks in as few turns as possible. Based on the manual inspection, we notice that the majority of users tend to ask the models for more add-on features in the latter conversation turns.

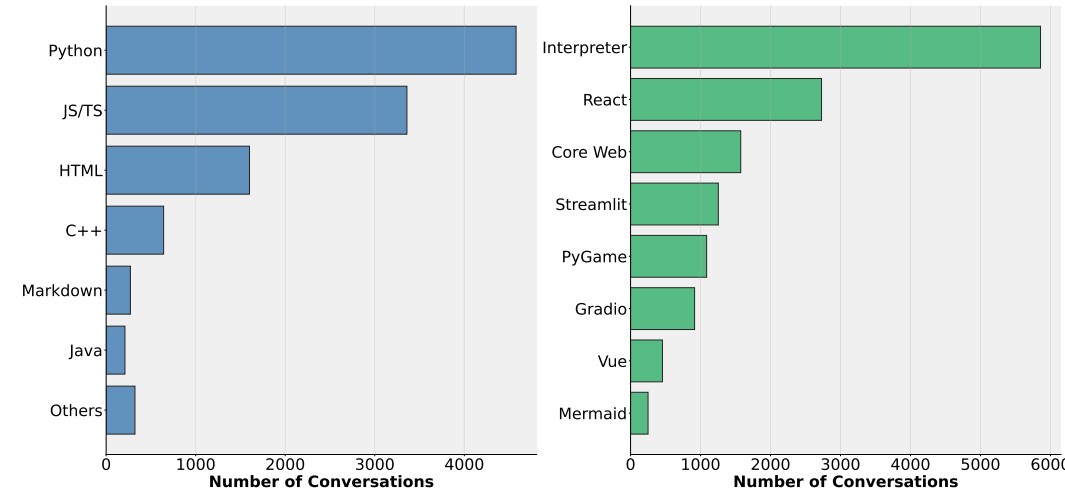

Figure 11: Distributions of languages (*left*) and frameworks (*right*) among the collected conversations.

**Languages and Frameworks**  As shown in Figure 11, Python dominates with more than 4,000 conversations, followed by JavaScript/TypeScript (3,359), HTML (1,601), and C++ (642). Smaller but non-negligible shares come from Markdown, Java, and an "Others" category that aggregates Go, Rust, and C. On the framework side, Interpreter sessions are most prevalent with nearly 6,000 conversations, reflecting heavy reliance on direct execution environments (primarily Python interpreters). React appears most frequently among frameworks (2,729), with Core Web (1,574), Streamlit (1,254), PyGame (1,087), and Gradio (915) also widely used, while Vue and Mermaid are less common. Overall, the distributions suggest that BIGCODEARENA usage is dominated by Python-centric and interpreter-based workflows, with a substantial portion targeting interactive or UI-oriented frameworks.

**Topic Modeling**  To better understand the diversity of prompts, we attempt to use the automatic topic modeling pipeline that was introduced in Chiang et al. (2024). However, the results do not show clear boundaries among each topic. To conclude reasonable programming topics, four of the authors manually inspect 50% of randomly sampled user prompts and identify six topics (with examples shown in Figure 2): (1) Web Design, building and styling websites, (2) Game Development, creating interactive games, (3) Diagram Creation, designing visual representations of systems or ideas, (4) Creative Coding, using code for artistic and experimental projects, and (5) Problem Solving, applying logical thinking to find efficient solutions.

F.2    UNDERSTANDING USER INTERACTION WITH EXECUTION OUTCOMES

**Observation Space**  For web-based programming frameworks covered by BIGCODEARENA, the observation space consists of a complete rendering of the interactive UI exposed through an iFrame, reflecting the output and side effects of executed code. This includes all visible changes to the page, user interface elements, and any interactive outcomes resulting from the program's execution. In alignment with prior research on agent-based interaction with web environments (Xie et al., 2024), BIGCODEARENA supports programmatic introspection via DOM access and event handling, enabling agents to perceive and reason about the structure and state of the interface. These raw observations enable rich interaction with dynamic and stateful web and application environments, but also present challenges in long-horizon reasoning and decision-making from high-resolution visual contexts and deeply nested DOM structures.

**Action Space**  The action space in BIGCODEARENA (Table 3) bypasses traditional browser sandbox constraints by directly recording user interactions with the rendered Web UI. Specifically, we capture screen resize events, mouse clicks, scroll up/down gestures, and keyboard inputs. Since the user can dynamically resize the browser window or viewport, all interactions are recorded using relative coordinates with respect to the displayed screen at each time step. Every user action is timestamped,

Table 3: Examples of the mouse and keyboard actions in BIGCODEARENA.

| Function | Description |
|---|---|
| click(x, y) | Perform a mouse click at screen coordinates $(x, y)$ |
| keyboard('enter') | Sends an Enter/Return key input |
| keyboard('x') | Sends a character key input (e.g., typing "x") |
| scroll(x) | Scroll up within $x$ units on the interface |
| resize(x, y) | Adjusts the window size to width $x$ and height $y$ |

allowing us to construct a precise and sequential interaction trajectory. This design enables rich logging of real-world usage patterns in a way that supports high-fidelity replay and learning from demonstrations, while avoiding limitations found in previous environments that restrict or abstract away low-level interaction signals. Examples can be seen in Section E.7.

Table 4: User interaction statistics across different sandbox environments in BIGCODEARENA

| Environment | # Sessions | # Keyboard | # Click | Duration (s) |
|---|---|---|---|---|
| React | 3,107 | 12.0 | 6.2 | 32.3 |
| Core Web | 1,699 | 18.8 | 6.3 | 59.6 |
| PyGame | 964 | 21.4 | 5.4 | 32.0 |
| Vue | 201 | 21.0 | 6.0 | 29.9 |
| Streamlit | 45 | 8.2 | 6.0 | 45.8 |
| Gradio | 14 | 13.7 | 4.2 | 26.1 |

**Distribution Analysis**  We analyze the distribution of UI interactions across 5,557 recorded sessions in BIGCODEARENA. The interaction patterns reveal distinct usage characteristics across different development environments and time scales. The majority of UI interactions are brief, with 72.0% (4,003 sessions) lasting 30 seconds or less, suggesting that users frequently engage in quick testing and validation cycles. Medium-duration interactions (30-120 seconds) account for 22.5% (1,249 sessions), while extended interactions exceeding 120 seconds comprise only 5.5% (305 sessions) of the dataset. Table 4 presents the interaction statistics across different sandbox environments. React dominates with 3,107 sessions (55.9%), followed by Core Web (30.6%) and PyGame (17.3%). The data reveals environment-specific interaction patterns: Core Web sessions exhibit the longest average interaction time (59.6 seconds) and highest keyboard event density (18.8 events/session), suggesting more text-heavy development. In contrast, Vue and Gradio sessions show shorter interaction times (29.9 and 26.1 seconds respectively), indicating more rapid prototyping cycles.

## G  BIGCODEARENA RANKING ANALYSIS

### G.1  OVERALL ANALYSIS

Analyzing our leaderboard across 4.7K multi-turn sessions involving 10 models (see Figure 3), we observe a consistent stratification of model performance across three evaluation settings: (1) All Data, (2) Environment Matched, and (3) Language Matched. These settings reflect progressively stricter controls to isolate confounding factors in model evaluation. The *All Data* setting includes all pairwise comparisons collected in our evaluation, regardless of the runtime environment or language in the response. As we notice that some users may ask a more generic question without specifying the languages or frameworks, we further introduce language-level and environment-level controls to disentangle model performance from such implicit variability and better reflect real-world deployment conditions. The *Environment Matched* setting restricts evaluation to comparisons in which both models were executed within the same sandbox environment, ensuring fairness with respect to system-level behavior such as resource allocation, file access, or execution speed. The *Language Matched* setting further narrows the evaluation scope to only those comparisons in which both models received prompts in the same natural language, controlling for potential discrepancies in multilingual handling or translation quality.

Across all three settings, we observe a stable and interpretable ranking structure. A clear top tier consistently emerges, led by o3-mini and o1-mini, achieving the highest Elo ratings with tight

confidence intervals. These models maintain the best performance regardless of environmental or linguistic constraints, showing robustness and broad applicability across coding scenarios. Just below them, Claude-3.5-Sonnet also performs strongly, narrowing the gap with the leaders in the language-matched setting. The next tier includes models such as GPT-4o, o1, and Gemini-2.0-Pro/Flash, whose rankings remain competitive but exhibit modest sensitivity to evaluation context. For example, GPT-4o shows slightly reduced performance in the language-matched condition, suggesting room for improvement in multilingual consistency. In contrast, Qwen2.5 models and Llama-3.3-70B consistently underperform across nearly all conditions, indicating a gap between frontier models and open alternatives.

## G.2 ANALYSIS OF PROGRAMMING TOPICS

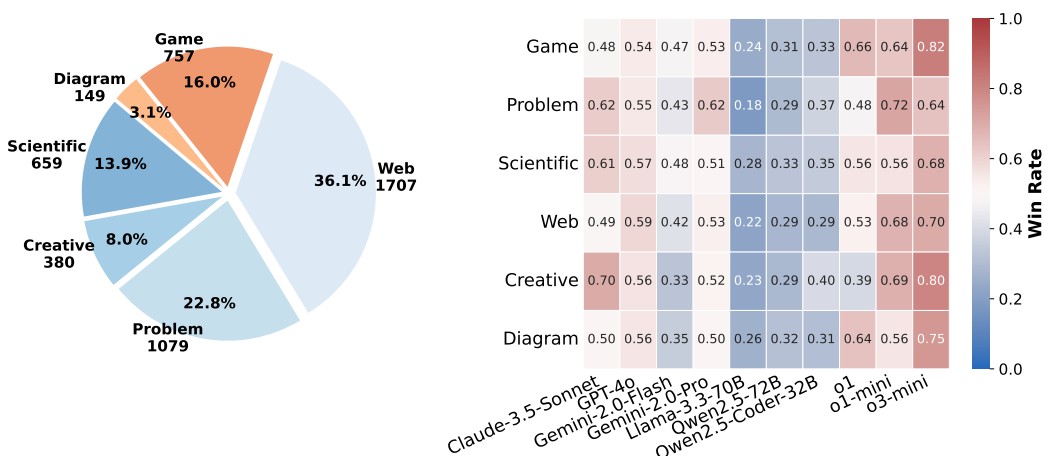

Figure 12: Distribution of programming topics (*left*) and model win rates across topics (*right*).

To further dissect model capabilities, we examine performance across distinct programming topics introduced in Section 3. We then use GPT-4.1-mini to classify the initial user prompt of each conversation into one of the six topics. The authors further check the classification results to confirm the quality. The distribution of prompts (Figure 12) reveals that web-related tasks dominate at 36.1%, followed by problem solving (22.8%), game development (16.0%), scientific computing (13.9%), creative coding (8.0%), and diagram generation (3.1%). This distribution reflects a strong emphasis on applied and interactive coding scenarios, consistent with real-world developer workloads. When comparing win rates across models segmented by topic, we observe clear performance stratification. Models such as o3-mini, o1-mini, and o1 consistently outperform others across all categories, achieving particularly high win rates in game development, problem-solving, and web-related tasks. Claude-3.5-Sonnet also demonstrates strong results, especially in creative coding, while maintaining competitive performance in scientific and problem-solving tasks. In contrast, Gemini-2.0-Pro and Gemini-2.0-Flash occupy a middle tier, without clear topic-specific dominance. Larger Qwen2.5 variants and Llama-3.3-70B lag significantly across most categories, with pronounced weaknesses in web and problem-solving prompts. These results underscore that while top models generalize broadly across domains, others show uneven strengths, and aggregate Elo scores can obscure important topic-specific differences.

## G.3 COMPARISONS TO PREVIOUS EVALUATIONS

To compare BIGCODEARENA with existing benchmarks, we use Spearman rank correlations to measure alignment across leaderboards. Figure 13 reveals that BIGCODEARENA is most aligned with Chatbot Arena, which is expected given their shared conversational format and reliance on large-scale user voting. Within Chatbot Arena, the coding-specific subset (Chatbot Arena-Coding) shows the strongest alignment with BIGCODEARENA ($\rho = 0.68$), since it isolates coding-related queries from general conversational ones. In contrast, BigCodeBench shows weaker alignment ($\rho = 0.43$), reflecting its restriction to Python-only benchmarks that fail to capture the broader diversity of coding

Figure 13: Spearman correlations and ranking shifts between BIGCODEARENA and other benchmarks, including Copilot Arena (Chi et al.), BigCodeBench (Zhuo et al.), Chatbot Arena (Chiang et al., 2024), Chatbot Arena (Coding), and WebDev Arena (lma, 2025b).

tasks. WebDev Arena also exhibits only moderate correlation with BIGCODEARENA ($\rho = 0.50$), likely due to its narrow emphasis on Next.js development, which biases the evaluation toward a limited slice of web technologies. However, when we compare rankings between the Core Web category of BIGCODEARENA with WebDev Arena, they are more aligned ($\rho = 0.68$), suggesting that BIGCODEARENA can cover more holistic code generation evaluation beyond the categories like web design.

### G.4 VALIDATION OF VOTE QUALITY

To assess the quality of crowdsourced votes, we randomly selected 470 sessions from the 4.7K multi-turn pairwise conversations and asked two human experts having more than 5 years of programming experience in covered languages and frameworks to relabel the label their preference per comparison. Similar to Chiang et al. (2024), the experts are only given the conversations blindly, and asked to execute all the generated code snippets and carefully interact with the execution results. The experts are required to vote the preference within 10 minutes. After analyzing the relabelling results, we notice high agreement rates between the original BIGCODEARENA annotators and two human experts. Specially, we find that the experts have the agreements of 80.4% and 86.0% with the original preference. We further measure the Kappa coefficient based on these relabelling statistics and obtain 0.61 and 0.72, indicating substantial agreement between the original annotators and the expert relabelling. We also measure the inter-annotator agreement between the two experts, finding an agreement rate of 83.2% and a corresponding Kappa coefficient of 0.67, suggesting strong consistency between the expert judgements. The remaining disagreements mainly due to the interactive nature of the evaluation. In many cases, both candidate solutions may run successfully but differ in how they handle inputs, error messages, or user interaction flows. Depending on how an expert interprets or engages with this execution feedback, different preferences can naturally emerge.

## H    ARTIFACTS

Table 5: Artifacts for reproducibility.

| Name | Public Link or Endpoint |
|---|---|
| *Evaluated Models in* BIGCODEARENA | |
| o3-mini | `o3-mini-2025-01-31` |
| o1-mini | `o1-mini-2024-09-12` |
| Claude-3.5-Sonnet | `claude-3-5-sonnet-20241022` |
| GPT-4o | `gpt-4o-2024-11-20` |
| o1 | `o1-2024-12-17` |
| Gemini-2.0-Pro | `gemini-2.0-pro-exp-02-05` |
| Gemini-2.0-Flash | `gemini-2.0-flash-exp` |
| Qwen2.5-Coder-32B | https://huggingface.co/Qwen/Qwen2.5-Coder-32B-Instruct |
| Qwen2.5-72B | https://huggingface.co/Qwen/Qwen2.5-72B-Instruct |
| Llama-3.3-70B | https://huggingface.co/meta-llama/Llama-3.3-70B-Instruct |
| *Models for Evaluations in* BIGCODEREWARD | |
| Claude-Sonnet-4 | `claude-sonnet-4-20250514` |
| Claude-3.7-Sonnet | `claude-3-7-sonnet-20250219` |
| Claude-3.5-Sonnet | `claude-3-5-sonnet-20241022` |
| GPT-4.1 | `gpt-4.1-2025-04-14` |
| GPT-4.1-mini | `gpt-4o-mini-2024-07-18` |
| GPT-4o | `gpt-4o-2024-11-20` |
| GPT-4o-mini | `gpt-4o-mini-2024-07-18` |
| Gemma-3-27B | https://huggingface.co/google/gemma-3-27b-it |
| Qwen2.5-VL-72B-Instruct | https://huggingface.co/Qwen/Qwen2.5-VL-72B-Instruct |
| Qwen2.5-VL-32B-Instruct | https://huggingface.co/Qwen/Qwen2.5-VL-32B-Instruct |
| InternVL3-78B | https://huggingface.co/OpenGVLab/InternVL3-78B |
| InternVL3-38B | https://huggingface.co/OpenGVLab/InternVL3-78B |
| GLM-4.5V | https://huggingface.co/zai-org/GLM-4.5V |
| MiMo-VL-7B-RL | https://huggingface.co/XiaomiMiMo/MiMo-VL-7B-RL |
| Kimi-VL-A3B-Thinking | https://huggingface.co/moonshotai/Kimi-VL-A3B-Thinking-2506 |
| *Evaluated Models in* AUTOCODEARENA | |
| GPT-5 | `gpt-5-2025-08-07` |
| Claude-Opus-4 | `claude-opus-4-20250514` |
| Claude-Sonnet-4 | `claude-sonnet-4-20250514` |
| Kimi-K2 | https://huggingface.co/moonshotai/Kimi-K2-Instruct |
| Gemini-2.5-Pro | `gemini-2.5-pro` |
| Qwen3-Coder | https://huggingface.co/Qwen/Qwen3-Coder-480B-A35B-Instruct |
| GLM-4.5 | https://huggingface.co/zai-org/GLM-4.5 |
| DeepSeek-V3.1 | https://huggingface.co/deepseek-ai/DeepSeek-V3.1 |
| GPT-4.1 | `gpt-4.1-2025-04-14` |
| DeepSeek-R1 | https://huggingface.co/deepseek-ai/DeepSeek-R1-0528 |
| GPT-OSS-120B | https://huggingface.co/openai/gpt-oss-120b |
| DeepSeek-V3 | https://huggingface.co/deepseek-ai/DeepSeek-V3-0324 |
| o4-mini | `o4-mini-2025-04-16` |
| GPT-OSS-20B | https://huggingface.co/openai/gpt-oss-20b |
| Claude-3.5-Sonnet | `claude-3-5-sonnet-20241022` |
| o3-mini | `o3-mini-2025-01-31` |
| Gemini-2.5-Flash | `gemini-2.5-flash` |
| Grok-Code | `grok-code-fast-1` |
| Claude-3.5-Haiku | `claude-3-5-haiku-20241022` |
| GPT-4o | `gpt-4o-2024-11-20` |
| GPT-4o-mini | `gpt-4o-mini-2024-07-18` |

# I    BIGCODEREWARD

## I.1    EXPERIMENT DETAILS

Since the input length for reward models can be long, the model sometimes fails to produce outputs in valid JSON format, which prevents correct parsing. In these cases, we use GPT-4.1-mini with the prompt below to reconstruct the model output into valid JSON.

```
The following is a response from a judge model that should be in JSON ←↩
    format, but it's not properly formatted. Please convert it to the ←↩
    required JSON format. "reasoning" is a single paragraph explanation←↩
     without line breaks. Any quotation marks within the text should be←↩
     properly escaped for a valid JSON format.

The expected JSON format should be:
{{
    "Overall": {{
        "winner": "A" | "B" | "TIE",
        "reasoning": "explanation for the overall judgment"
    }},
}}

Original response:
{original_response}

Please output ONLY the JSON format, no additional text or explanation.
"""
```

## I.2    JUDGEMENT PROMPT (WITH OUTPUT)

```
You are a code-review judge assigned to compare two candidate ←↩
    solutions (A and B) against a user's programming request. Your job ←↩
    is to evaluate each submission and choose an overall winner based ←↩
    on how well each solution implements the requested features.

Evaluation Criteria:
Your primary focus should be: The solution implements every requested ←↩
    feature accurately and correctly without adding or omitting ←↩
    functionality. Consider multiple aspects, including code efficiency←↩
    , explanation, readability, maintainability, correctness, and UI/UX←↩
    , but the most critical factor is the complete and accurate ←↩
    implementation of all requested features.

Winner Options:
- "A": Solution A is clearly better
- "B": Solution B is clearly better
- "Tie": Both solutions are roughly equivalent in quality

Evaluation Process:
You should evaluate based on the combination of:
- The code implementation
- Code output or results produced
- Visual rendering results
- How completely each solution address the original request

Input Format:
<|Instruction|>
{INSTRUCTION}

<|The Start of Assistant A's Answer|>
<|The Start of Code|>
{code_A}
<|The End of Code|>
```

```
<|The Start of Execution Results|>
Output: {sandbox_output}
Error: {sandbox_error}
<|The End of Execution Results|>
<|The Start of Assistant A's Artifact Screenshot|>
{SCREENSHOT_A}
<|The End of Assistant A's Artifact Screenshot|>
<|The End of Assistant A's Answer|>

<|The Start of Assistant B's Answer|>
<|The Start of Code|>
{code_B}
<|The End of Code|>
<|The Start of Execution Results|>
Output: {sandbox_output}
Error: {sandbox_error}
<|The End of Execution Results|>
<|The Start of Assistant B's Artifact Screenshot|>
{SCREENSHOT_A}
<|The End of Assistant B's Artifact Screenshot|>
<|The End of Assistant B's Answer|>

Output Format:
Return exactly one JSON object with this schema below. "reasoning" is ←
    a single paragraph explanation without line breaks. Any quotation ←
    marks within the text should be properly escaped for a valid JSON ←
    format.
```json
{
 "Overall": {
   "winner": "A"|"B"|"Tie",
   "reasoning": "..."
 }
}
```
```

## I.3 JUDGEMENT PROMPT (WITHOUT OUTPUT)

```
You are a code-review judge assigned to compare two candidate ←
    solutions (A and B) against a user's programming request. Your job ←
    is to evaluate each submission and choose an overall winner based ←
    on how well each solution implements the requested features.

Important: You will only see the code implementations, not their ←
    execution results or screenshots. Focus your evaluation purely on ←
    code quality, structure, and theoretical correctness.

Evaluation Criteria:
Your primary focus should be: The solution implements every requested ←
    feature accurately and correctly without adding or omitting ←
    functionality. Consider multiple aspects, including code efficiency←
    , explanation, readability, maintainability, correctness, and UI/UX←
    , but the most critical factor is the complete and accurate ←
    implementation of all requested features.

Winner Options:
- "A": Solution A is clearly better
- "B": Solution B is clearly better
- "Tie": Both solutions are roughly equivalent in quality

Evaluation Process:
```

```
You should evaluate based on:
- The code implementation
- How completely each solution address the original request

Input Format:
<|Instruction|>
{INSTRUCTION}

<|The Start of Assistant A's Answer|>
<|The Start of Code|>
{code_A}
<|The End of Code|>
<|The End of Assistant A's Answer|>

<|The Start of Assistant B's Answer|>
<|The Start of Code|>
{code_B}
<|The End of Code|>
<|The End of Assistant B's Answer|>

Output Format
Return exactly one JSON object with this schema:
```json
{
 "Overall": {
   "winner": "A"|"B"|"Tie",
   "reasoning": "..."
 }
}
```

### I.4 METRIC

We evaluate models using accuracy and macro F1. The label space is $\{\texttt{A}, \texttt{B}, \texttt{Tie}\}$, where the reward judge decides whether model $A$ is preferred, model $B$ is preferred, or the outputs are equally good.

**Accuracy.** Accuracy is defined as the proportion of predictions that exactly match the ground-truth label:

$$\text{Accuracy} = \frac{\#\{\text{correct predictions}\}}{\#\{\text{all examples}\}}.$$

**Macro F1.** For each class $c \in \{\texttt{A}, \texttt{B}, \texttt{Tie}\}$, we compute precision, recall, and F1:

$$\text{Precision}_c = \frac{\text{TP}_c}{\text{TP}_c + \text{FP}_c}, \quad \text{Recall}_c = \frac{\text{TP}_c}{\text{TP}_c + \text{FN}_c},$$

$$\text{F1}_c = \frac{2 \cdot \text{Precision}_c \cdot \text{Recall}_c}{\text{Precision}_c + \text{Recall}_c}.$$

The macro F1 is the average across the three classes:

$$\text{Macro F1} = \frac{1}{3} \sum_{c \in \{\texttt{A},\texttt{B},\texttt{Tie}\}} \text{F1}_c.$$

### I.5 EXPERIMENT RESULTS

Table 6: Macro F1 scores (%) for reward models across different task categories with and without execution outputs. "–" denotes evaluation without execution outputs, "+" denotes evaluation with execution outputs. Best results in each category are highlighted in bold.

| Models | Web | | Game | | Creative | | Diagram | | Scientific | | Problem | | Overall | |
|---|---|---|---|---|---|---|---|---|---|---|---|---|---|---|
| | – | + | – | + | – | + | – | + | – | + | – | + | – | + |
| *Proprietary Models* | | | | | | | | | | | | | | |
| Claude-4-Sonnet | 46.0 | 47.7 | 42.6 | 48.2 | 45.2 | 47.0 | 42.2 | 62.7 | 39.0 | 46.6 | 41.5 | 49.2 | 43.4 | 48.9 |
| Claude-3.7-Sonnet | **49.9** | 54.6 | 43.1 | 49.1 | **52.7** | **56.3** | 42.3 | 65.3 | 41.4 | 48.6 | 41.4 | 51.8 | 46.1 | 53.2 |
| Claude-3.5-Sonnet | 48.2 | 47.7 | 43.0 | 45.9 | 50.4 | 50.5 | 39.2 | 48.2 | 42.5 | 46.0 | 48.0 | 52.8 | 46.7 | 48.9 |
| GPT-4.1 | 47.5 | 51.0 | **45.7** | 52.0 | 47.2 | 50.0 | 42.2 | 60.9 | 42.2 | 48.7 | 40.0 | 46.2 | 45.0 | 50.1 |
| GPT-4.1-mini | 45.9 | 49.0 | 42.3 | 47.7 | 44.0 | 49.9 | 34.8 | 56.7 | 39.2 | 50.3 | 41.0 | 48.6 | 43.4 | 49.6 |
| GPT-4o | 48.5 | 52.9 | 45.8 | 49.5 | 52.4 | 52.4 | **46.3** | 60.4 | 43.0 | 52.9 | 40.1 | 49.1 | 46.2 | 52.1 |
| GPT-4o-mini | 42.4 | 49.5 | 42.4 | 46.4 | 44.2 | 49.4 | 38.8 | 62.5 | 42.1 | 51.1 | 47.5 | 52.1 | 44.0 | 50.7 |
| *Open Models* | | | | | | | | | | | | | | |
| Gemma-3-27B | 43.6 | 48.2 | 43.9 | 46.8 | 48.0 | 48.2 | 40.8 | 64.7 | 40.7 | 49.9 | **49.5** | 52.8 | 45.6 | 50.6 |
| Qwen2.5-VL-72B Instruct | 49.8 | **55.5** | 44.3 | **53.9** | 51.4 | 55.7 | 43.3 | **68.8** | **45.7** | 56.9 | 48.1 | **57.1** | **48.9** | **56.8** |
| Qwen2.5-VL-32B-Instruct | 45.7 | 48.2 | 40.9 | 47.4 | 44.1 | 55.4 | 40.1 | 57.5 | 42.3 | **57.9** | 48.4 | 54.1 | 45.5 | 52.4 |
| InternVL3-78B | 47.9 | 35.3 | 42.7 | 35.5 | 48.8 | 36.6 | 38.3 | 34.3 | 44.1 | 40.3 | 45.4 | 48.3 | 46.7 | 39.6 |
| InternVL3-38B | 44.1 | 34.7 | 42.0 | 33.9 | 45.2 | 33.4 | 40.0 | 34.8 | 40.0 | 42.4 | 46.9 | 50.3 | 44.8 | 39.5 |
| GLM-4.5V | 46.9 | 49.0 | 42.7 | 44.6 | 49.0 | 48.8 | 41.2 | 50.9 | 43.0 | 46.4 | 44.2 | 46.2 | 45.6 | 47.5 |
| MiMo-VL-7B-RL | 42.2 | 43.0 | 41.6 | 43.7 | 43.6 | 48.0 | 49.4 | 53.3 | 39.6 | 47.3 | 38.6 | 40.4 | 42.1 | 44.3 |
| Kimi-VL-A3B-Thinking | 39.8 | 40.7 | 37.4 | 40.9 | 40.0 | 46.5 | 33.7 | 50.0 | 38.2 | 43.0 | 36.5 | 36.5 | 38.4 | 41.2 |

## I.6 CASE STUDIES

### I.6.1 WEB DESIGN

```html
1  <!DOCTYPE html>
2  <html lang="en">
3  # <... omitted head/style code ...>
4  <body>
5      <header>
6          <h1>Fitness Studio</h1>
7          <nav>
8              <a href="#classes">Classes</a>
9              # <... omitted table ...>
10         </nav>
11     </header>
12     <section id="classes" class="class-schedule">
13         <h2>Class Schedules</h2>
14         <table>
15             <thead>
16                 <tr>
17                     <th>Class</th>
18                     # <... omitted table ...>
19                 </tr>
20             </thead>
21             <tbody>
22                 <tr>
23                     <td>Yoga</td>
24                     <td>6:00 AM - 7:00 AM</td>
25                     <td>Jane Doe</td>
26                     <td><button ⤶
     onclick="openBookingModal('Yoga')">Book</button></td>
27                 </tr>
28             # <... omitted table ...>
29
30             </tbody>
31         </table>
32     </section>
33 # <... omitted schedule section...>
34     </div>
35     <script>
36         # <... omitted js code logic ...>
37     </script>
38 </body>
39 </html>
40
```

```html
1  <!DOCTYPE html>
2  <html lang="en">
3  <head>
4  # <... omitted head/style code ...>
5  </head>
6  <body>
7    <header>
8      <h1>Welcome to Our Fitness Studio</h1>
9    </header>
10   <nav>
11     <a href="#schedule">Class Schedules</a>
12     <a href="#trainers">Trainer Bios</a>
13     # <... omitted table ...>
14   </nav>
15   <div class="container">
16     <section id="schedule">
17       <h2>Class Schedules</h2>
18       <p>Check out our weekly class schedules ⤶
     below:</p>
19       <ul>
20         <li>Monday: Yoga - 7 AM, HIIT - 6 PM</li>
21         <li>Tuesday: Pilates - 7 AM, Zumba - ⤶
     6 PM</li>
22         # <... omitted table ...>
23       </ul>
24     </section>
25     # <... omitted booking section code ...>
26   </div>
27   <footer>
28     <p>© 2023 Fitness Studio. All Rights ⤶
     Reserved.</p>
29   </footer>
30   <script>
31 # <... omitted js code ...>
32   </script>
33 </body>
34 </html>
35
```

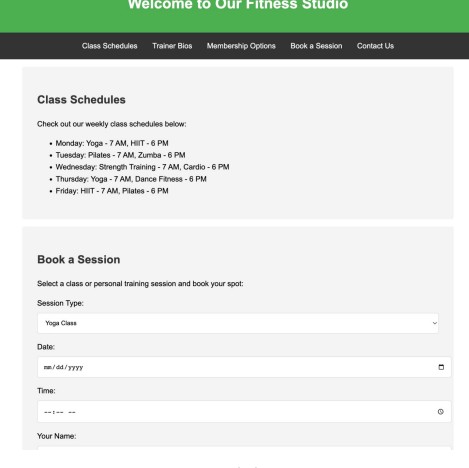

Model A

Model B

| Condition | Human | InternVL3 -38B | InternVL3 -78B | MiMo-VL -7B-RL | Gemma-3 -27B | GLM -4.5V | GPT -4.1 | GPT-4.1 -mini | GPT -4o | GPT-4o -mini | Kimi-VL -A3B-Thinking | Qwen2.5-VL -32B-Instruct | Qwen2.5-VL -72B-Instruct | Claude-3.5 -Sonnet | Claude-3.7 -Sonnet | Claude-4 -Sonnet |
|---|---|---|---|---|---|---|---|---|---|---|---|---|---|---|---|---|
| With Output | **A** | A | B | A | A | A | A | A | A | A | A | A | A | A | A | A |
| Without Output | **A** | A | A | A | A | A | A | A | A | A | A | A | A | A | A | A |

(a) Reward Model Voting Results. A: Vote for Model A    B: Vote for Model B    Tie: Vote Tie

```
1  import React, { useState, useEffect } from
       'react';
2  const LightSequencer = () => {
3    const [lights, setLights] = useState(() =>
         Array(25).fill(false));
4    const [isPlaying, setIsPlaying] =
         useState(false);
5    const [currentStep, setCurrentStep] =
         useState(0);
6    const [savedPatterns, setSavedPatterns] =
         useState([]);
7    const [patternName, setPatternName] =
         useState('');
8    const toggleLight = (index) => {
9      const newLights = [...lights];
10     newLights[index] = !newLights[index];
11     setLights(newLights);
12   };
13   const playSequence = () => # <... omitted ...>
14   useEffect(() => # <... omitted ...>
15   const savePattern = () => # <... omitted ...>
16   const loadPattern = (pattern) => # <...
         omitted ...>
17   return (
18     <div className="flex flex-col
         items-center mt-8">
19       <div className="grid grid-cols-5 gap-2">
20         # <... omitted ...>
21       </div>
22       <div className="mt-4 flex flex-col
         items-center">
23         # <... omitted ...>
24         <button
25           className="px-4 py-2 bg-blue-500
         text-white rounded-lg # <... omitted
         ...> >{isPlaying ? 'Stop' : 'Play'}
26         </button>
27       </div>
28     </div>
29   );
30 };
31 export default LightSequencer;
32
```

```
1  import React, { useState, useEffect } from
       'react';
2  interface SavedPattern {
3    name: string;
4    grid: boolean[][];
5  }
6  const LightSequencer: React.FC = () => {
7    const [grid, setGrid] = useState<boolean[][]>(
8      Array.from({ length: 5 }, () =>
         Array(5).fill(false))
9    );
10   const [isPlaying, setIsPlaying] =
         useState(false);
11   const [speed, setSpeed] = useState(500);
12   const [currentStep, setCurrentStep] =
         useState(0);
13   const [savedPatterns, setSavedPatterns] =
         useState<SavedPattern[]>([]);
14   const [patternName, setPatternName] =
         useState('');
15   const [showSaveModal, setShowSaveModal] =
         useState(false);
16   const [showLoadModal, setShowLoadModal] =
         useState(false);
17   // Save patterns to localStorage whenever
         savedPatterns changes
18   useEffect(() => # <... omitted ...>
19   const toggleLight = (row: number, col:
         number) => # <... omitted ...>
20   const resetGrid = () => {# <... omitted ...>
21   const savePattern = () => # <... omitted ...>
22   const loadPattern = (pattern: SavedPattern)
         => {
23     setGrid(pattern.grid);
24     setShowLoadModal(false);
25   };
26   const deletePattern = (patternToDelete:
         SavedPattern) => # <... omitted ...>
27   return (# <... omitted ...>);
28 };
29 export default LightSequencer;
30
```

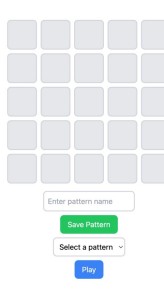

Model A

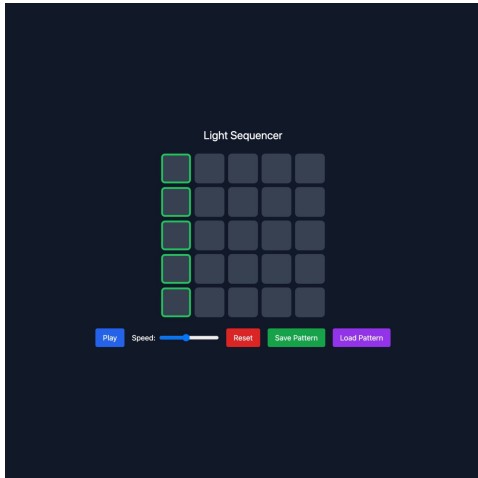

Model B

| Condition | Human | InternVL3 -38B | InternVL3 -78B | MiMo-VL -7B-RL | Gemma-3 -27B | GLM -4.5V | GPT -4.1 | GPT-4.1 -mini | GPT -4o | GPT-4o -mini | Kimi-VL -A3B-Thinking | Qwen2.5-VL -32B-Instruct | Qwen2.5-VL -72B-Instruct | Claude-3.5 -Sonnet | Claude-3.7 -Sonnet | Claude-4 -Sonnet |
|---|---|---|---|---|---|---|---|---|---|---|---|---|---|---|---|---|
| With Output | **B** | Tie | A | B | B | B | B | B | B | B | – | B | B | B | B | B |
| Without Output | **B** | B | B | – | B | B | B | B | B | B | A | B | B | B | B | B |

(a) Reward Model Voting Results. A: Vote for Model A    B: Vote for Model B    Tie: Vote Tie

### I.6.3 CREATIVE CODING

```python
import streamlit as st
import matplotlib.pyplot as plt
import matplotlib.image as mpimg
import numpy as np
from scipy.ndimage import gaussian_filter
def apply_blur(image, blur_level, texture):
    # <... omitted ...>
def main():
    st.title("Frosted Glass Blur Effect ↵
    Simulator")
    # Upload image
    image_file = st.file_uploader("Upload an ↵
    image", type=["jpg", "png"])
    if image_file is not None:
        image = mpimg.imread(image_file)
        # Display original image
        st.header("Original Image")
        fig, ax = plt.subplots()
        ax.imshow(image)
        ax.axis('off')
        st.pyplot(fig)
        # Adjust blur level / Select texture ↵
    / Apply blur effect
        # <... omitted ...>
        # Display blurred image
        st.header("Blurred Image")
        fig, ax = plt.subplots()
        ax.imshow(blurred_image)
        ax.axis('off')
        st.pyplot(fig)
if __name__ == "__main__":
    main()
```

```html
<!DOCTYPE html>
<html lang="en">
<head>
    # <... omitted ...>
</head>
<body>
    <h1>Frosted Glass Effect Simulator</h1>
    <div class="scene-container">
    # <... omitted table ...>
    </div>
    <div class="controls">
        <div class="control-group">
            <label for="blurSlider">Blur ↵
    Intensity: 0px</label>
            <input type="range" ↵
    id="blurSlider" min="0" max="20" value="0">
        </div>
        # <... omitted table ...>
        <div class="control-group"># <... ↵
    omitted ...></div>
    </div>
    <script>
        // Generate random colors
        function getRandomColor() {# <... ↵
    omitted ...>}
        // Create background pattern
        function createBackground() {# <... ↵
    omitted ...>}
        // Change texture
        function changeTexture(textureType) ↵
    {# <... omitted ...>}
    </script>
</body>
</html>
```

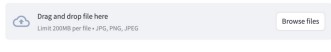

Model A

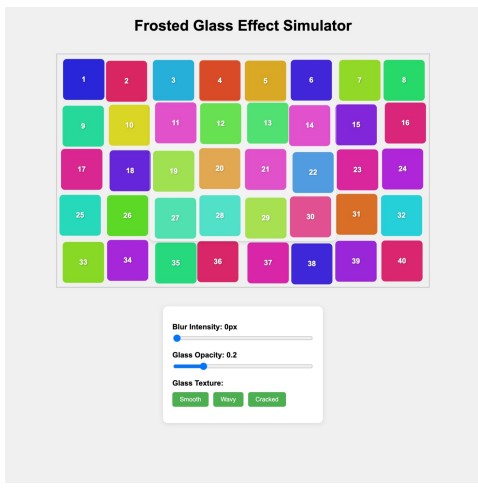

Model B

| Condition | Human | InternVL3 -38B | InternVL3 -78B | MiMo-VL -7B-RL | Gemma-3 -27B | GLM -4.5V | GPT -4.1 | GPT-4.1 -mini | GPT -4o | GPT-4o -mini | Kimi-VL -A3B-Thinking | Qwen2.5-VL -32B-Instruct | Qwen2.5-VL -72B-Instruct | Claude-3.5 -Sonnet | Claude-3.7 -Sonnet | Claude-4 -Sonnet |
|---|---|---|---|---|---|---|---|---|---|---|---|---|---|---|---|---|
| With Output | **B** | Tie | B | B | B | B | B | B | B | B | B | A | B | B | B | B |
| Without Output | **B** | B | B | B | B | B | B | B | B | A | A | A | B | B | B | B |

(a) Reward Model Voting Results. A: Vote for Model A    B: Vote for Model B    Tie: Vote Tie

### I.6.4 DIAGRAM CREATION

```
1  graph TD;
2  %% Styling for clarity
3  classDef decisionPoint
          fill:#FFD700,stroke:#333,stroke-width:2,color:#000,sh
4  classDef outcome
          fill:#87CEEB,stroke:#333,stroke-width:2,color:#000,sh
5
6  Start["Start: Evaluate Investment
          Options"]:::decisionPoint -->
          RiskAssessment["Risk
          Assessment"]:::decisionPoint;
7
8  # <... omitted ...>
9  MarketConditionsLow -->|Stable|
          InvestLow["Decision: Invest in Low-Risk
          Option"]:::outcome;
10 MarketConditionsLow -->|Unstable|
          Reconsider["Decision: Reconsider
          Options"]:::outcome;
11
12 MarketConditionsMedium -->|Growing|
          InvestMedium["Decision: Invest in
          Medium-Risk Option"]:::outcome;
13 MarketConditionsMedium -->|Stagnant| Reconsider;
14
15 MarketConditionsHigh -->|High Volatility|
          InvestHigh["Decision: Invest in
          High-Risk Option"]:::outcome;
16 MarketConditionsHigh -->|Too Volatile|
          Reconsider;
17
18 Reconsider --> Start["Go Back to
          Start"]:::outcome;
19
```

```
1  graph TD
2  Start[Start Investment Decision]
3  RiskAssessment{Risk Assessment}
4  ExpectedReturn{Expected Return}
5  InvestmentDuration{Investment Duration}
6  MarketConditions{Market Conditions}
7
8  Start --> RiskAssessment
9
10 RiskAssessment -->|Low Risk| ExpectedReturnLow
11 RiskAssessment -->|Medium Risk|
          ExpectedReturnMedium
12 RiskAssessment -->|High Risk| ExpectedReturnHigh
13 # <... omitted ...>
14 Conservative -->|Dashed Line|
          EndConservative[End: Conservative
          Investment Option]
15 Balanced -->|Dashed Line| EndBalanced[End:
          Balanced Investment Option]
16 Aggressive -->|Dashed Line|
          EndAggressive[End: Aggressive
          Investment Option]
17
18 Conservative --> Bonds
19 Conservative --> RealEstate
20
21 Balanced --> Stocks
22 Balanced --> Bonds
23 Balanced --> RealEstate
24
25 Aggressive --> Stocks
26 Aggressive --> Crypto
27 Aggressive --> RealEstate
28
```

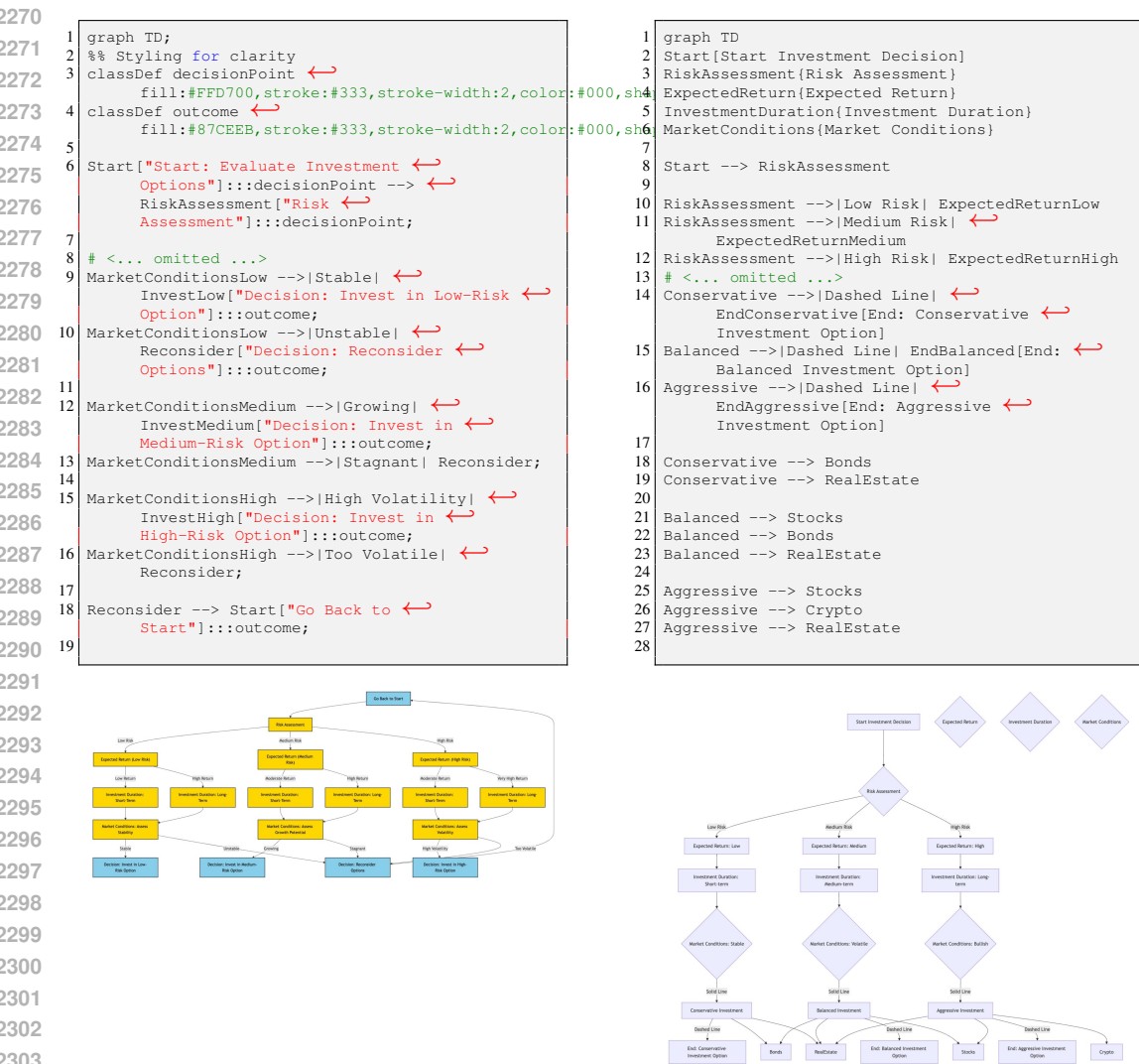

Model A                                        Model B

| Condition | Human | InternVL3-38B | InternVL3-78B | MiMo-VL-7B-RL | Gemma-3-27B | GLM-4.5V | GPT-4.1 | GPT-4.1-mini | GPT-4o | GPT-4o-mini | Kimi-VL-A3B-Thinking | Qwen2.5-VL-32B-Instruct | Qwen2.5-VL-72B-Instruct | Claude-3.5-Sonnet | Claude-3.7-Sonnet | Claude-4-Sonnet |
|---|---|---|---|---|---|---|---|---|---|---|---|---|---|---|---|---|
| With Output | **A** | B | A | A | A | A | A | A | A | A | A | A | A | A | A | A |
| Without Output | **A** | A | A | A | A | A | A | A | A | A | A | A | A | A | A | A |

(a) Reward Model Voting Results. A: Vote for Model A    B: Vote for Model B

### I.6.5 SCIENTIFIC COMPUTING

```python
import gradio as gr
import numpy as np, random
from PIL import Image

def diffusion_limited_aggregation(grid_size=50,
        num_particles=500, stickiness=0.1):
    """Simulates Diffusion-Limited
    Aggregation (DLA) on a 2D grid."""
    grid = np.zeros((grid_size, grid_size),
        dtype=int)
    grid[grid_size//2, grid_size//2] = 1  #
        seed particle

    for _ in range(num_particles):
        x, y = random.randint(0,
        grid_size-1), random.randint(0, grid_size-1)
        while True:
            dx, dy =
        random.choice([-1,1,0,0]),
        random.choice([0,0,-1,1])
            x, y = (x+dx) % grid_size, (y+dy)
        % grid_size
                # <... omitted random walk &
        sticking logic ...>
                break
    img = Image.fromarray(np.uint8(grid *
        255)).resize((300,300), Image.NEAREST)
    return img

iface = gr.Interface(
    fn=diffusion_limited_aggregation,
    inputs=[gr.Slider(20,100,50,1,"Grid Size"),

        gr.Slider(100,1000,500,10,"Particles"),

        gr.Slider(0.01,1.0,0.1,0.01,"Stickiness")],
    outputs=gr.Image(label="DLA Cluster"),
    title="Diffusion-Limited Aggregation
        Visualizer",
    description="Simulates fractal growth
        with adjustable parameters."
)
iface.launch()
```

```python
import gradio as gr, numpy as np, random,
    matplotlib.pyplot as plt

class DiffusionLimitedAggregation:
    def __init__(self, grid_size=50,
    num_particles=500):
        self.grid_size, self.num_particles =
    grid_size, num_particles
        self.grid = np.zeros((grid_size,
    grid_size), dtype=int)
        self.grid[grid_size//2, grid_size//2]
    = 1  # central seed

    def simulate(self):
        # <... omitted particle spawning,
    random walk & sticking logic ...>
        return self.grid

def run_dla_simulation(grid_size, num_particles):
    # <... omitted input validation &
    simulation call ...>

    plt.imshow(DiffusionLimitedAggregation(grid_size,
    num_particles).simulate(),
            cmap='viridis'); return plt.gcf()

def create_dla_app():
    with gr.Blocks() as demo:
        gr.Markdown("# Diffusion-Limited
    Aggregation Simulator")
        grid_size =
    gr.Slider(20,100,50,1,"Grid Size")
        num_particles =
    gr.Slider(100,1000,500,1,"Number of
    Particles")
        gr.Button("Run Simulation").click(
            fn=run_dla_simulation,
    inputs=[grid_size,num_particles],
            outputs=gr.Plot(label="DLA
    Cluster Visualization"))
    return demo

if __name__ == "__main__":
    create_dla_app().launch()
```

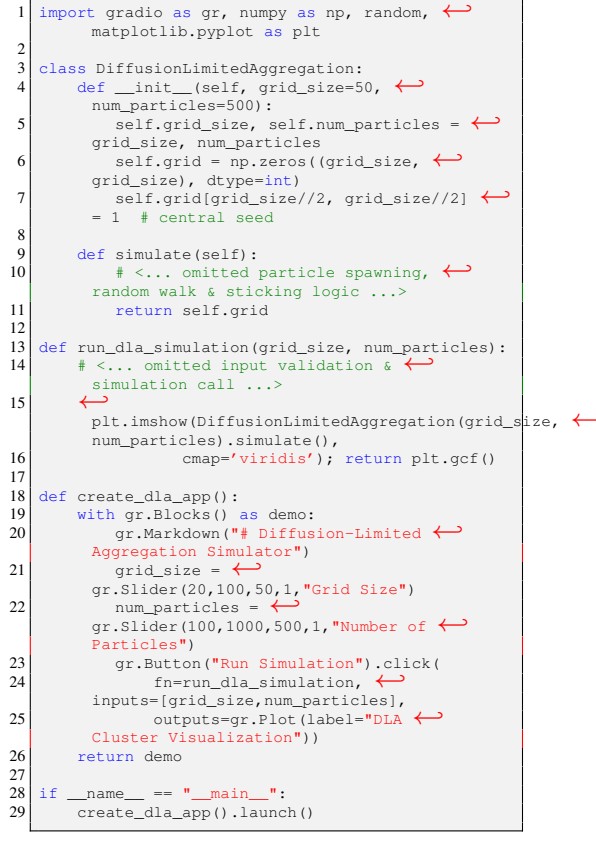

Model A                                    Model B

| Condition | Human | InternVL3 -38B | InternVL3 -78B | MiMo-VL -7B-RL | Gemma-3 -27B | GLM -4.5V | GPT -4.1 | GPT-4.1 -mini | GPT -4o | GPT-4o -mini | Kimi-VL -A3B-Thinking | Qwen2.5-VL -32B-Instruct | Qwen2.5-VL -72B-Instruct | Claude-3.5 -Sonnet | Claude-3.7 -Sonnet | Claude-4 -Sonnet |
|---|---|---|---|---|---|---|---|---|---|---|---|---|---|---|---|---|
| With Output | B | A | A | B | A | B | A | A | A | B | B | A | B | A | A | A |
| Without Output | B | A | B | B | A | B | A | A | B | B | B | B | B | B | A | B |

(a) Reward Model Voting Results. A: Vote for Model A    B: Vote for Model B    Tie: Vote Tie

### I.6.6 PROBLEM SOLVING

```python
import gradio as gr
import numpy as np
import heapq

class AStarPathfinder:
    def __init__(self, grid_size=10):
        self.grid_size = grid_size
        self.grid = np.zeros((grid_size, ←
          grid_size), dtype=int)
    def heuristic(self, a, b):
        # <... omitted ...>
    def get_neighbors(self, current):
        # <... omitted ...>
    def a_star(self, start, goal, ←
      preset_obstacles=None):
        # <... omitted ...>
    def visualize_path(self, path, grid):
        # <... omitted ...>

# Predefined Example Scenarios
EXAMPLE_SCENARIOS = [# <... omitted ...>]

def find_path(scenario_index, start_x=None, ←
      start_y=None, goal_x=None, goal_y=None):
    # <... omitted ...>
    return visualization

# Create Gradio interface
iface = gr.Interface(
    # <... omitted ...>
)
# Dynamic input visibility based on scenario ←
      selection
def update_inputs(scenario):
    # <... omitted ...>
iface.config["dependencies"][0]["targets"] = ←
      [0, 1, 2, 3, 4]
iface.config["dependencies"][0]["updates"] = ←
      update_inputs
if __name__ == "__main__":
    iface.launch()
```

```python
import gradio as gr
import heapq
import numpy as np
def heuristic(a, b):
    # <... omitted ...>
def a_star_search(grid, start, end):
    """Perform A* search on the grid to find ←
      the shortest path."""
    # <... omitted ...>
# Gradio app logic
def shortest_path(grid_input, start, end):
    """Find the shortest path using A* ←
      algorithm and return the grid with the ←
      path."""
    # <... omitted ...>

example_grid = [[0, 0, 1, 0],
                [0, 1, 0, 0],
                [0, 0, 0, 1],
                [1, 0, 0, 0]]

example_start = (0, 0)
example_end = (3, 3)

# Gradio interface
description = """
### A* Shortest Path Finder
Provide a grid (2D list) where '0' represents ←
      walkable cells and '1' represents obstacles.
# <... omitted ...>
#### Example Inputs:
- **Grid**: `[[0, 0, 1, 0], [0, 1, 0, 0], [0, ←
      0, 0, 1], [1, 0, 0, 0]]`
- **Start**: `(0, 0)`
- **End**: `(3, 3)`
"""
demo = gr.Interface(
    # <... omitted ...>
)
if __name__ == "__main__":
    demo.launch()
```

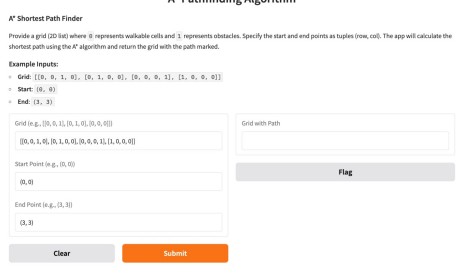

Model A

Model B

| Condition | Human | InternVL3 -38B | InternVL3 -78B | MiMo-VL -7B-RL | Gemma-3 -27B | GLM -4.5V | GPT -4.1 | GPT-4.1 -mini | GPT -4o | GPT-4o -mini | Kimi-VL -A3B-Thinking | Qwen2.5-VL -32B-Instruct | Qwen2.5-VL -72B-Instruct | Claude-3.5 -Sonnet | Claude-3.7 -Sonnet | Claude-4 -Sonnet |
|---|---|---|---|---|---|---|---|---|---|---|---|---|---|---|---|---|
| With Output | B | A | – | A | A | A | A | A | A | A | A | A | A | A | A | A |
| Without Output | B | A | A | A | A | A | A | A | A | A | A | A | A | A | A | A |

(a) Reward Model Voting Results. A: Vote for Model A    B: Vote for Model B    Tie: Vote Tie    –: No response

# J  AUTOCODEARENA

## J.1  CLASSIFICATION PROMPT

```
You are a classification expert tasked with categorizing programming ↩
    instructions. Given a user instruction for a code model, classify ↩
    it into one of the following 6 categories:

Categories:

1. system programming
    - Security & encryption
    - Cloud computing
    - DevOps
    - Database

2. scientific computing
    - Data processing & cleaning
    - Data visualization & plotting
    - Scientific/numeric programming
    - Statistical analysis & modeling
    - Machine learning algorithms
    - Deep learning implementations

3. algorithmic programming
    - competitive programming
    - Data structures (arrays, trees, graphs, etc.)
    - General programming concepts & syntax
    - Language-specific problems

4. web design
    - web-based application
    - webpage development

5. creative coding
    - SVG art
    - Visual art
    - Design-focused coding tasks

6. game development
    - Game logic implementation
    - Game mechanics

Instructions:
- Read the user instruction carefully
- Choose the single most appropriate category
- If the instruction spans multiple categories, choose the ↩
     primary/dominant one
- Output your result in JSON format

User Instruction to Classify:
[INSERT INSTRUCTION HERE]

Output Format:
```json
{
  "category_id": [number],
  "category_name": "[category name]",
}
```

## J.2  GENERATION PROMPT

The prompt is the same as the one shown in Section E.5.

## J.3 JUDGEMENT SYSTEM PROMPT

```
Please act as an impartial judge and evaluate the quality of the code ↵
    provided by two AI assistants to the user prompt. You will be ↵
    given assistant A's answer and assistant B's answer, along with ↵
    the execution results of their code. Your job is to evaluate which ↵
    assistant's generated code is better.

When evaluating the assistants' answers, compare both assistants'code ↵
    execution results (e.g., stdout, stderr, and screenshot of the ↵
    rendered code) first. You must identify and correct any mistakes ↵
    or inaccurate information.

Note that the stderr may contain warnings only and you must not take ↵
    it as an error. Due to the limitation of the execution ↵
    environment, the errors may not be due to the code itself but the ↵
    incompatibility issues or the lack of dependencies. These should ↵
    be considered when evaluating the code.

There are several cases for the side-by-side comparison:
- Case 1: Both assistants' code execution results are successful. If ↵
    screenshots are provided, you should compare the screenshots of ↵
    the rendered code.
- Case 2: One assistant's code execution results are successful, ↵
    while the other's are not. If the failure of the assistant's code ↵
    execution results is due to the limitation of the execution ↵
    environment, you MUST NOT penalize the assistant's response. You ↵
    MUST carefully check the code generated by the assistant and judge ↵
    the code correctness.
- Case 3: Both assistants' code execution results are not successful. ↵
    You should compare both assistants' responses only. You MUST ↵
    carefully check the code generated by the assistants and judge the ↵
    code correctness.

There are several scenarios for coding tasks:
- web design: the web page or application should be able to run in ↵
    the browser and the user should be able to see the result. UI and ↵
    UX are the most important factors.
- game development: the game should be able to run and the user ↵
    should be able to see the result. UI, UX, and the game logic are ↵
    the most important factors.
- creative coding: the artifact should produce a creative work. The ↵
    creativity and novelty are the most important factors.
- problem solving: the code should be able to solve the problem ↵
    described by the user. The correctness and efficiency are the most ↵
    important factors.
- scientific computing: the code should use the proper scientific ↵
    methods and tools to solve the problem. The correctness, ↵
    efficiency, and visualization are the most important factors.
- diagram creation: the code should be able to create a diagram for ↵
    logic or data flow. The visual presentation and the clarity are ↵
    the most important factors.

YOU MUST IGNORE THE FAILURES OF THE CODE EXECUTION RESULTS THAT ARE ↵
    DUE TO THE LIMITATION OF THE ENVIRONMENT. YOU MUST NOT JUDGE BASED ↵
    ON THE EXISTENCE OF TEST CASES GENERATED BY THE ASSISTANTS. IF ANY ↵
    SCREENSHOTS OR VISUAL OUTPUTS ARE PROVIDED, YOU MUST INSPECT THEM ↵
    CAREFULLY FIRST. IF YOU CANNOT TELL THE QUALITY OF THE CODE BASED ↵
    ON THE EXECUTION RESULTS, YOU SHOULD INSPECT THE CODE.

YOU MUST NOT TAKE THE COMPLEXITY OF THE SETUP PROCESS INTO ACCOUNT. ↵
    REQUIRING MORE DEPENDENCIES DOES NOT MEAN THAT THE CODE IS LESS ↵
    PREFERABLE. REMEMBER, THE OUTCOME IS MORE IMPORTANT THAN THE ↵
    PROCESS. DEPENDENCIES DO NOT MATTER.
```

```
24  THINK FROM THE USER'S PERSPECTIVE.
25
26  After providing your explanation, you must output only one of the ←
        following choices as your final verdict with a label:
27
28  1. Assistant A is significantly better: [[A>>B]]
29  2. Assistant A is slightly better: [[A>B]]
30  3. Tie, relatively the similar or hard to tell: [[A=B]]
31  4. Assistant B is slightly better: [[B>A]]
32  5. Assistant B is significantly better: [[B>>A]]
33
34  Example output: "My final verdict is tie: [[A=B]]".
```

## J.4  CUSTOMIZED SANDBOX

We customize the sandbox environment for AUTOCODEARENA, which is different from the original one used by BIGCODEARENA. The new execution pipeline introduces a fully local, Docker-backed system that replaces the E2B-based remote sandboxing model. This architectural shift eliminates dependence on external control planes and ensures deterministic, inspectable behavior suitable for rigorous benchmarking. The primary difference lies in container lifecycle management designed for security and robustness. Each run creates a fresh container with strict resource limits, dropped capabilities, and a non-root sandbox user. Unlike the remote model that relies on managed VMs, our approach uses ephemeral host directories bind-mounted to container workspaces, enabling efficient artifact collection while preserving isolation. Command execution emphasizes reliability through thread-level timeouts that avoid daemon destabilization. Background servers are managed with proper process isolation and container-local logging, contrasting with the remote API-based process management of the original system. A thread-safe port allocator coordinates concurrent web workloads locally rather than through provider ingress. A central innovation is artifact- and visualization-centric observability. The pipeline injects lightweight instrumentation into Python code to intercept plotting library calls and force non-interactive backends, ensuring visual outputs are captured deterministically. After execution, the system performs directory diffs to classify new files by type with content-based deduplication, while simultaneously parsing stdout for embedded visual content. This dual approach ensures comprehensive visual capture across libraries and rendering modalities, replacing the executor-native result objects of the remote system. For interactive web applications, the pipeline provides uniform support across frameworks by writing applications to container workspaces, managing dependencies locally, and acquiring headless screenshots using embedded browsers from within the sandbox. This contrasts sharply with the original approach that exposes applications through external provider ingress and relies on remote screenshot capabilities. The integration of container hardening, filesystem-based artifact discovery, and in-sandbox headless introspection creates a self-contained execution substrate that treats visual evidence as first-class output. By bringing isolation and orchestration on-device, the new pipeline offers reproducibility, privacy, and transparent failure modes while scaling parallel evaluation through bounded creation and parallelized cleanup. These properties address the determinism, traceability, and isolation requirements paramount for automated assessment in AUTOCODEARENA.

## J.5 MORE RESULTS

### J.5.1 WEB DESIGN

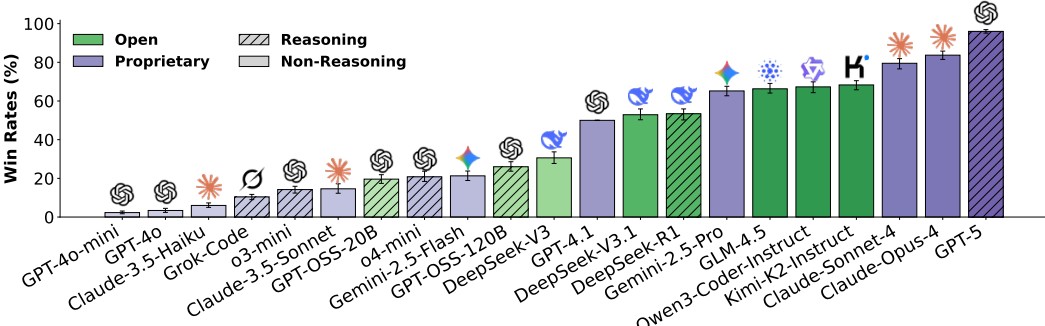

Figure 20: Web design performance of open and proprietary models on AUTOCODEARENA. We use GPT-4.1 as the baseline system and Claude-3.7-Sonnet as the judge. To avoid the potential judgement bias towards self-generated responses, we exclude Claude-3.7-Sonnet from the rankings.

### J.5.2 GAME DEVELOPMENT

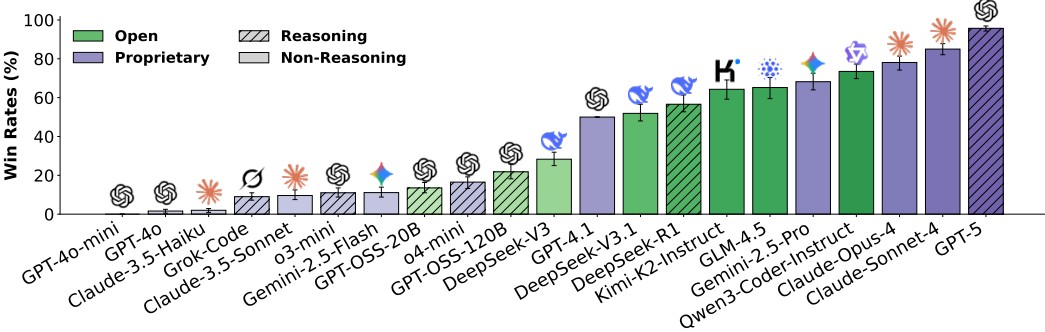

Figure 21: Game development performance of open and proprietary models on AUTOCODEARENA. We use GPT-4.1 as the baseline system and Claude-3.7-Sonnet as the judge. To avoid the potential judgement bias towards self-generated responses, we exclude Claude-3.7-Sonnet from the rankings.

### J.5.3 CREATIVE CODING

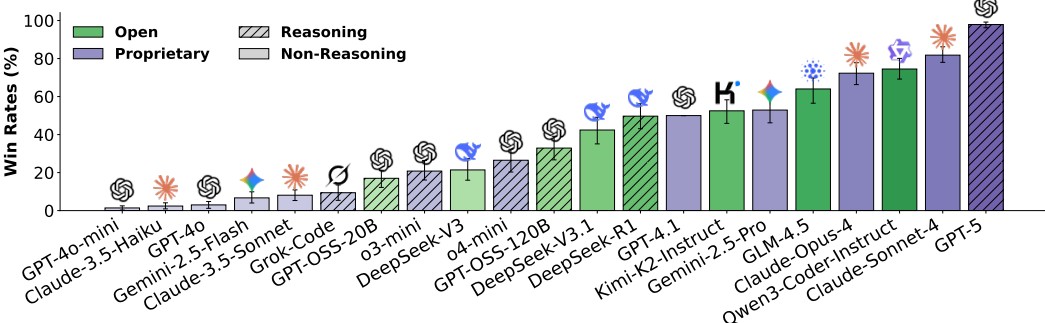

Figure 22: Creative coding performance of open and proprietary models on AUTOCODEARENA. We use GPT-4.1 as the baseline system and Claude-3.7-Sonnet as the judge. To avoid the potential judgement bias towards self-generated responses, we exclude Claude-3.7-Sonnet from the rankings.

### J.5.4 DIAGRAM CREATION

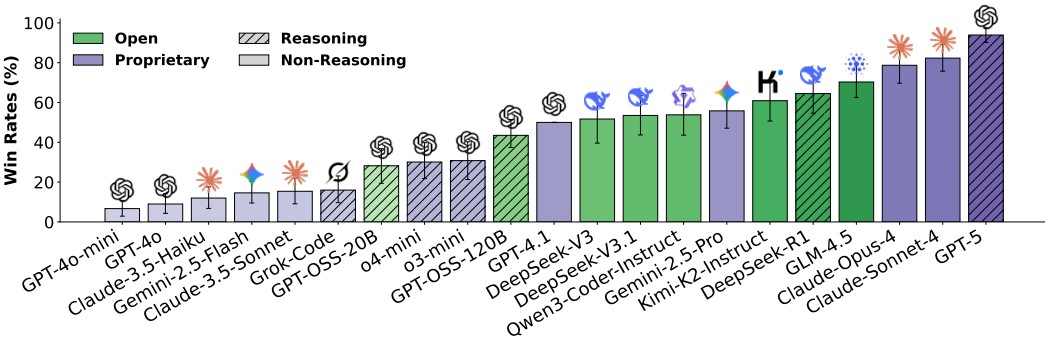

Figure 23: Diagram creation performance of open and proprietary models on AUTOCODEARENA. We use GPT-4.1 as the baseline system and Claude-3.7-Sonnet as the judge. To avoid the potential judgement bias towards self-generated responses, we exclude Claude-3.7-Sonnet from the rankings.

### J.5.5 SCIENTIFIC COMPUTING

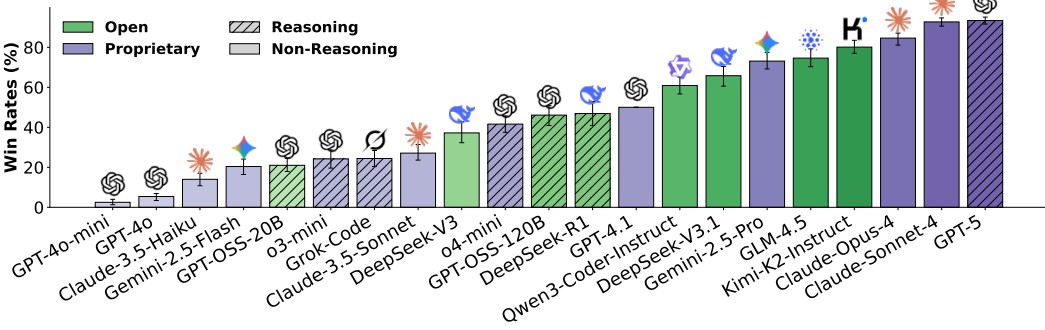

Figure 24: Scientific computing performance of open and proprietary models on AUTOCODEARENA. We use GPT-4.1 as the baseline system and Claude-3.7-Sonnet as the judge. To avoid the potential judgement bias towards self-generated responses, we exclude Claude-3.7-Sonnet from the rankings.

## J.5.6 PROBLEM SOLVING

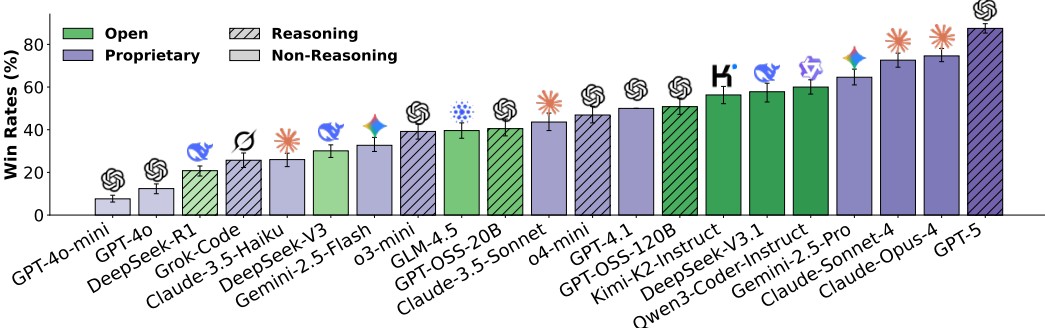

Figure 25: Problem solving performance of open and proprietary models on AUTOCODEARENA. We use GPT-4.1 as the baseline system and Claude-3.7-Sonnet as the judge. To avoid the potential judgement bias towards self-generated responses, we exclude Claude-3.7-Sonnet from the rankings.

## J.6 CASE STUDIES

| **Instructions** | **Model Outputs** |
|---|---|

Write an HTML page with embedded JavaScript that creates an interactive human anatomy diagram. Each body part (e.g., heart, lungs, brain) should: Display its name, function, and related diseases on hover. Allow filtering by system (e.g., circulatory, respiratory, nervous). Include a search bar to find body parts by name or function.

Claude-4-Opus

GPT-5

Gemini-2.5-Pro

Deepseek-V3.1

Qwen3-Coder

Kimi-K2

Simulate a fluid flow using the Navier-Stokes equations, visualizing the motion of particles in a liquid or gas medium. Use JS

Claude-4-Opus

GPT-5

Gemini-2.5-Pro

Deepseek-V3.1

Qwen3-Coder

Kimi-K2

Table 7: Model Comparisons – Queries 1 & 2

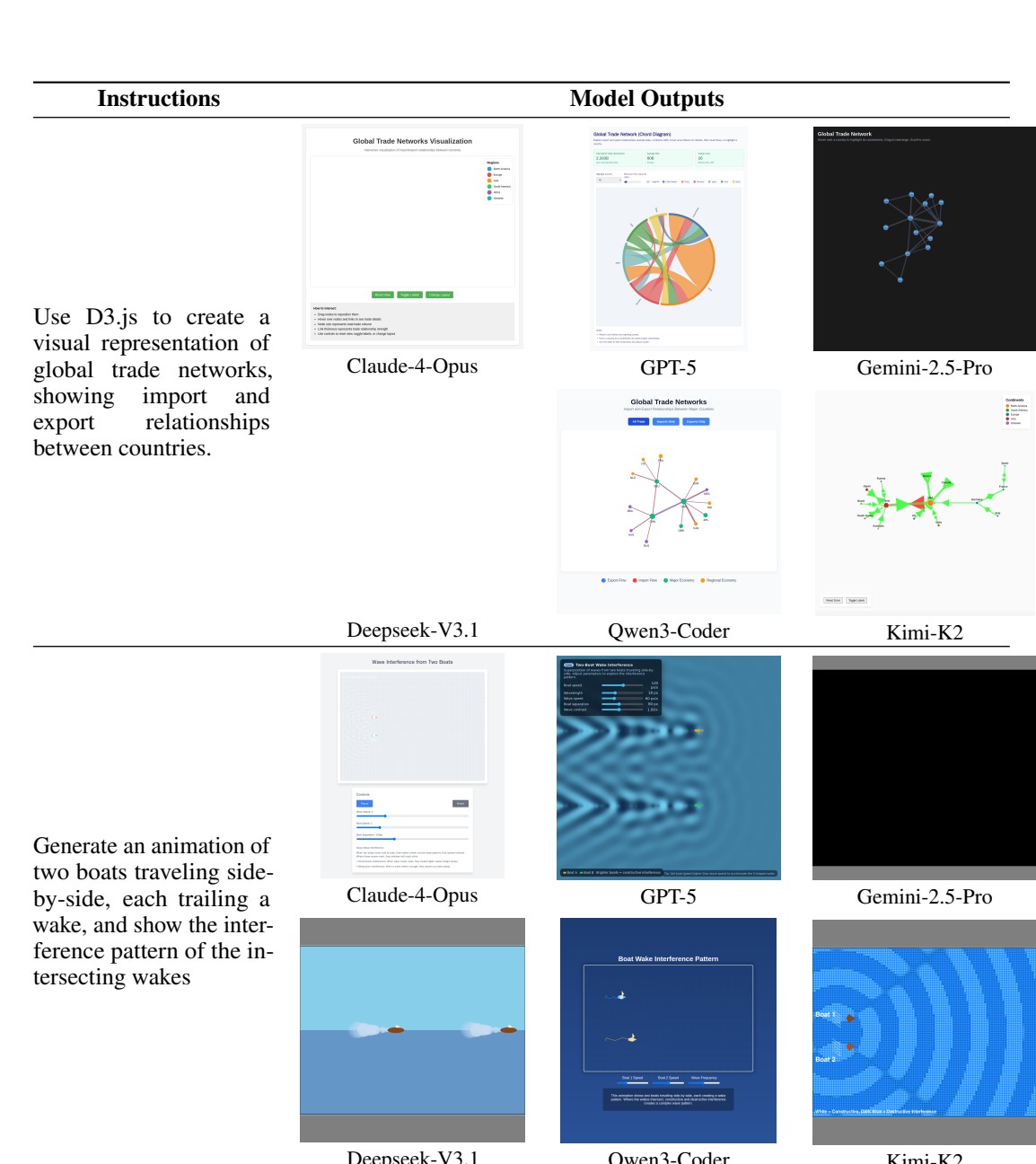

| Instructions | Model Outputs |
| --- | --- |

Table 8: Model Comparisons – Queries 3 & 4

| Instructions | Model Outputs |
| --- | --- |

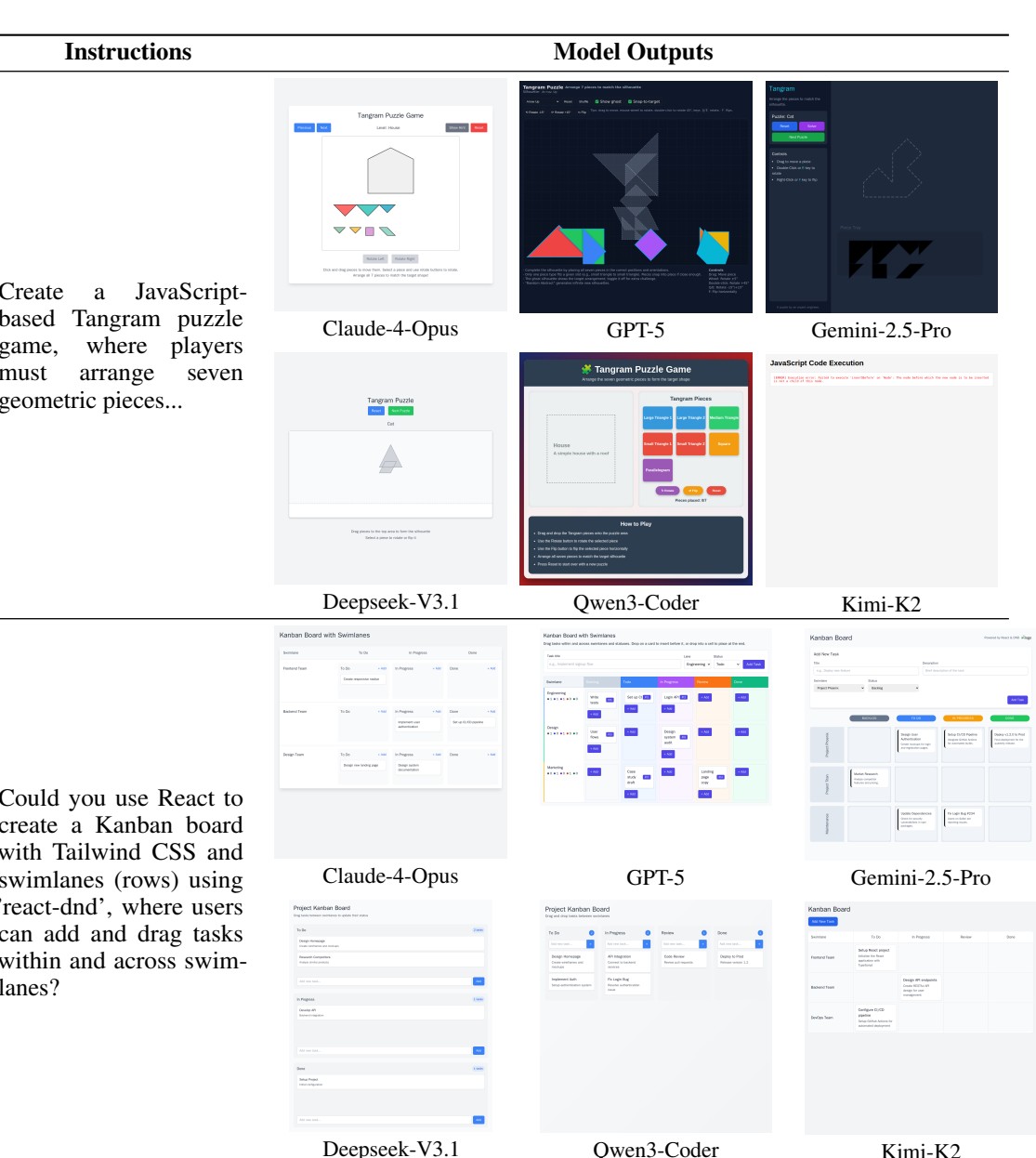

Create a JavaScript-based Tangram puzzle game, where players must arrange seven geometric pieces...

Could you use React to create a Kanban board with Tailwind CSS and swimlanes (rows) using 'react-dnd', where users can add and drag tasks within and across swimlanes?

Table 9: Model Comparisons – Queries 5 & 6

| **Instructions** | **Model Outputs** |
|---|---|
| Build a mobile app using Gradio for real-time weather forecasting. The app should allow users to view forecasts, receive alerts for severe weather, and track conditions in multiple locations. Provide an intuitive interface for navigation and personalization. Optimize the app for performance on mobile devices. | 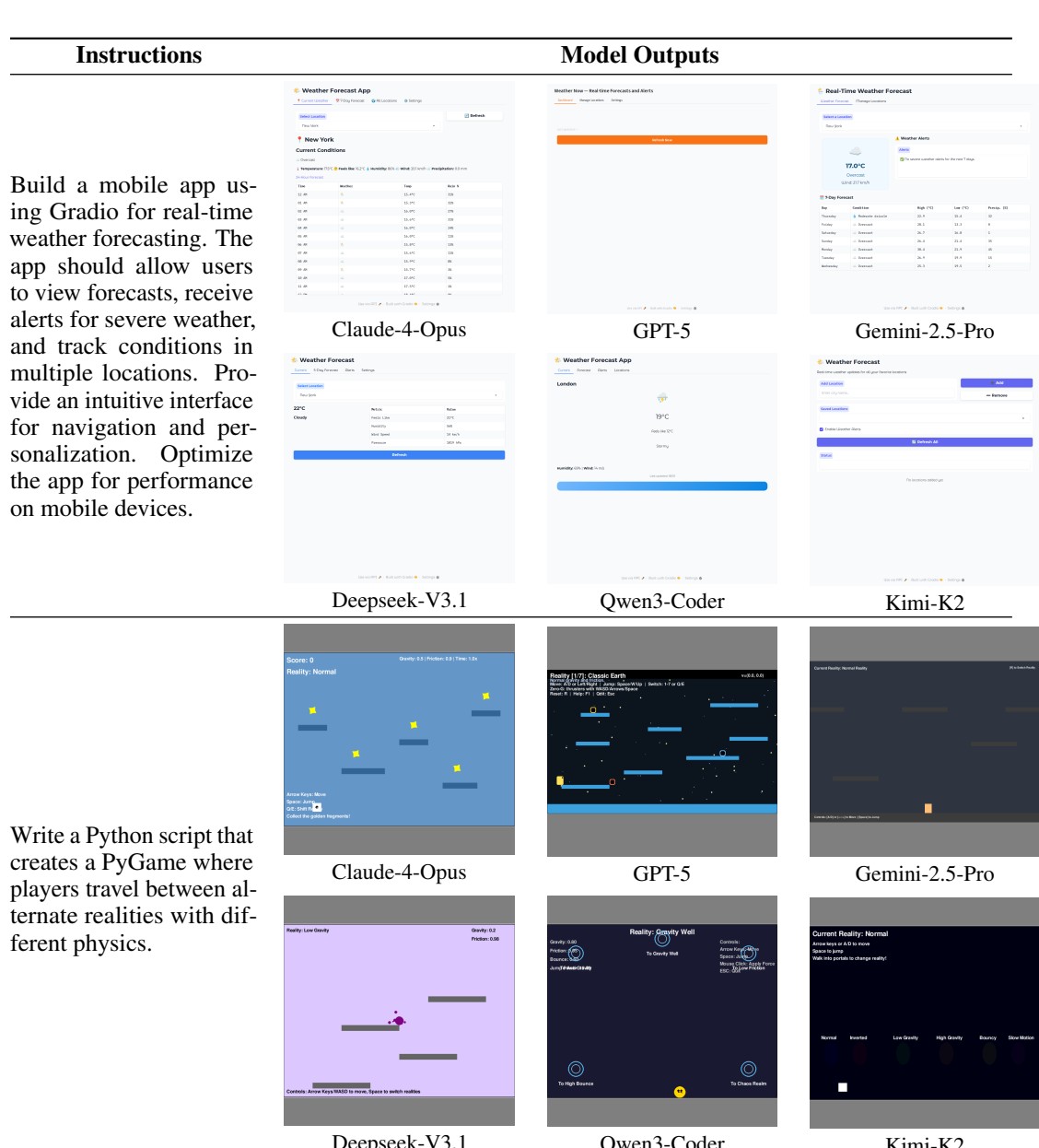 |

Table 10: Model Comparisons – Queries 7 & 8

| Instructions | Model Outputs |
|---|---|
| Using streamlit build an user friendly income tax calculator and explainer that determines an individual's estimated annual income tax based on their gross annual salary and US state location. Allow users to enter their gross annual salary and select their US state location. Then do the tax calculation and output the estimated annual income tax based on the user's input, taking into account federal and state income tax rates. | 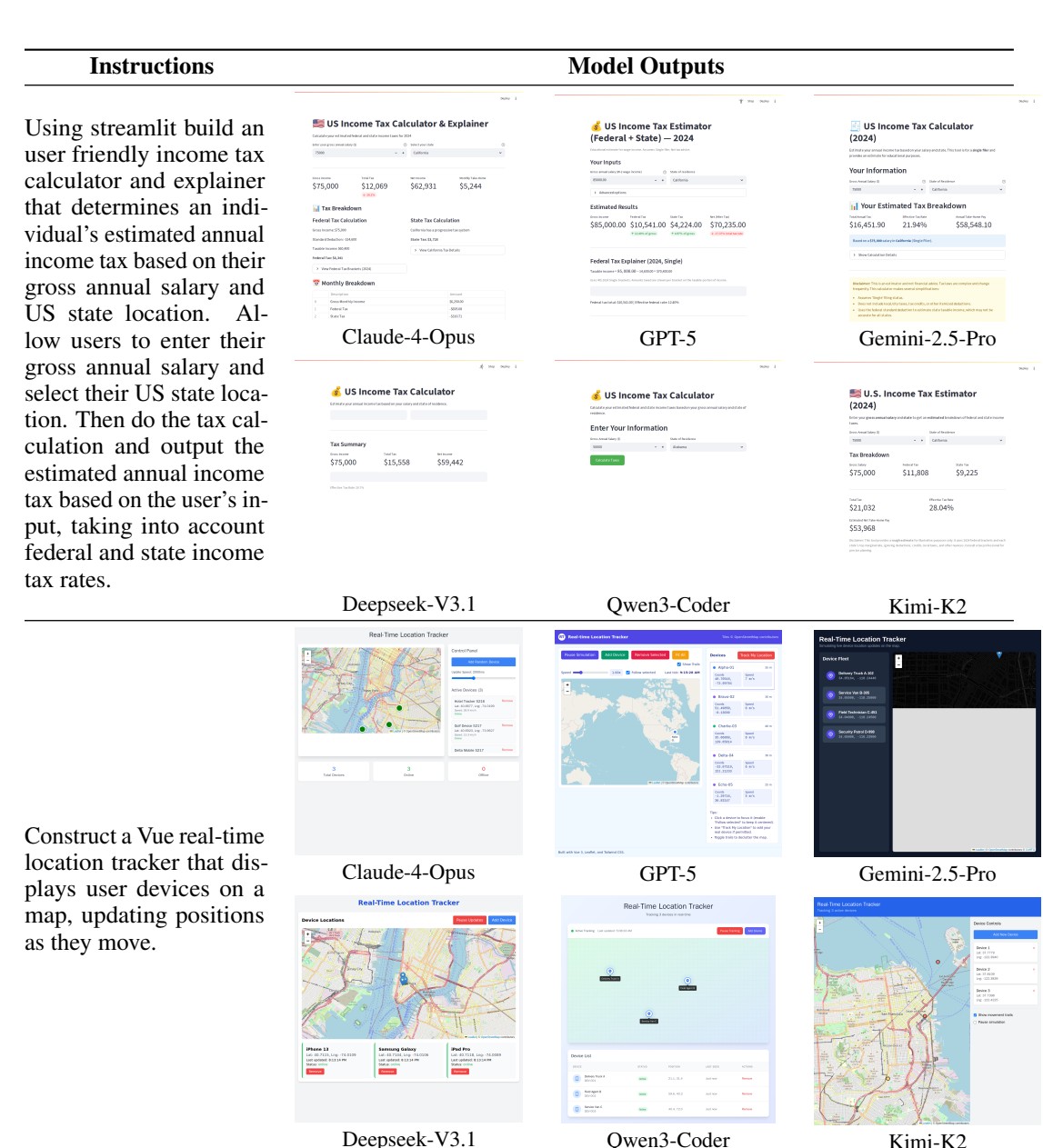 |

Table 11: Model Comparisons – Queries 9 & 10

