# OpenReview forum: "BigCodeArena: Unveiling More Reliable Human Preferences in Code Generation via Execution"
_ICLR.cc/2026/Conference — ICLR 2026 Conference Withdrawn Submission_

### Official Review · Reviewer_umg5 · 2025-10-26

**Soundness:** 4
**Presentation:** 4
**Contribution:** 2
**Rating:** 2
**Confidence:** 5

**Summary:**

This paper proposes an open crowdsourced evaluation platform for code generation that uses an on-the-fly execution environment.
This helps collecting more reliable human preference data, by mitigating errors from human judgment based only on static code review.

**Strengths:**

* Human preference collection: Execution feedback towardsmore robust human preference data.
* Extensive collection: 10 languages and 8 execution environments with interactive debugging.
* Benchmarks: BIGCODEREWARD and AUTOCODEARENA would help future research.

**Weaknesses:**

* Sensitivity to quality:  As its robustness heavily depends on the quality and consistency of human-provided feedback, more analysis to justify possible low-quality inputs from human (see questions below)
* Human cost: Collecting human preference data, especially across multiple languages and interactive environments can be expensive and time-consuming. Reporting time/infracost per each data point would be interesting
* Model train:  It remains unclear how effectively these signals could be integrated into training pipelines when sufficiently collected. Discussion whether this can be used for training better model would be useful.

**Questions:**

Any answer to weaknesses would help rebuttal
Specifically, how can we ensure reliability and consistency of human feedback?
How can we quantify or filter out low-quality feedbacks?
How sensitive are the evaluation metrics or model performance results to noisy feedbacks?
What is the average time and cost per annotation or preference collection, and whether automation can reduce cost.
How scalable to expand to new languages or execution contexts?

---

### Official Review · Reviewer_gpiC · 2025-10-31

**Soundness:** 4
**Presentation:** 2
**Contribution:** 4
**Rating:** 8
**Confidence:** 3

**Summary:**

This paper introduces BigCodeArena, an open human evaluation platform for code generation, featuring real-time execution and interactive debugging environments. The authors collected over 14K code-centric conversation sessions across 10 large language models, including 4.7K multi-turn sessions with pairwise human preference labels.
Building on this dataset, the paper further proposes two benchmarks: (i) BigCodeReward — evaluating the consistency between reward model judgments and human preferences over 4.7K annotated conversations; and (ii) AutoCodeArena — an automated Elo-based benchmark for assessing coding quality without human evaluators.
The study highlights that (1) proprietary and open-source LLMs show comparable reliability in judging code quality, (2) execution feedback substantially improves preference alignment, and (3) GPT-5 achieves the best overall code generation quality among recent models.

**Strengths:**

- The work establishes a robust and transparent data collection framework for execution-based human evaluation.
- The paper provides comprehensive design details on sandboxing, environment configuration, and preference aggregation, enhancing reproducibility.
- The proposed benchmarks (BigCodeReward, AutoCodeArena) form a meaningful step toward automated and scalable evaluation of code generation systems.

**Weaknesses:**

- The presentation could be improved for clarity and precision. Some parts, including figures and methodological descriptions, are confusing and would benefit from clearer exposition (see Questions).
- The analysis feels somewhat limited, particularly on the code generation side. While Elo-based comparisons are informative, a deeper breakdown of error types, execution failures, or qualitative behavior differences across models would strengthen the findings.

**Questions:**

- Section 3 states that annotators conducted "multi-turn conversations with at least two user–model exchanges,"" but it remains unclear whether these interactions occurred independently with each model or jointly within the pairwise evaluation.

- In Figure 3, the meanings of All Data, Environment Matched, and Language Matched are insufficiently specified. It is unclear whether these settings refer to pair-sampling constraints during evaluation (i.e., which model pairs are compared) or to post-hoc averaging/grouping criteria applied when aggregating Elo scores.

---

### Official Review · Reviewer_XV5J · 2025-10-31

**Soundness:** 3
**Presentation:** 3
**Contribution:** 2
**Rating:** 4
**Confidence:** 4

**Summary:**

-BigCodeArena builds on Chatbot Arena to collect a dataset of 4.7k preferences.
- From this data, the authors introduce BigCodeReward to evaluate the consistency between reward models and human preferences.
- Additionally, the authors also introduce AutoCodeArena to automate the process of preference collection.

**Strengths:**

- The paper tackles an important topic, addressing the need for reliable human evaluation of LLM-generated code.
- It is generally well-written and easy to follow.
- The authors demonstrate multiple uses of the collected data, showing how human preferences can power both BigCodeReward and AutoCodeArena.

**Weaknesses:**

- The paper lacks actionable insights or design recommendations for practitioners—either developers using coding assistants or researchers building LLMs—reducing its practical significance. Relatedly, Several components of the work appear incremental relative to prior platforms such as Chatbot Arena and WebDevArena; similarly, AutoCodeArena resembles prior automated evaluation setups like Arena-Hard. The paper could better articulate what unique insights BigCodeArena contributes beyond integrating existing ideas or infrastructure.
- It is unclear how BigCodeReward handles noisy or inconsistent human preferences, which could affect the reliability of results.
- I could not find analysis or results on the response rate or completion statistics for the optional sub-questions, leaving unclear how representative these annotations are.
- There is no comparison to existing leaderboards or ranking trends (e.g., Chatbot Arena), and the observation that frontier proprietary models remain strongest is unsurprising. At a cursory glance, there seems to be a strong correlation.

**Questions:**

Please address each of the weaknesses.

---

### Official Review · Reviewer_WBdo · 2025-11-02

**Soundness:** 2
**Presentation:** 4
**Contribution:** 2
**Rating:** 2
**Confidence:** 5

**Summary:**

This paper introduces BIGCODEARENA, an open and execution-based human evaluation platform for code generation models. Specifically, the platform is built upon Chatbot Arena, which integrates real-time code execution, interactive debugging, and pairwise preference collection to produce a more reliable measure of model performance. Moreover, the authors collect 14K code-centric conversations across 10 LLMs, 10 programming languages, and 8 execution environments, from which 4.7K multi-turn preference samples are curated.

**Strengths:**

1. The authors correctly identify a key limitation in existing human evaluation: humans often cannot judge code quality without execution. By integrating executable environments, this work provides a more realistic and reliable evaluation protocol.
2. The writing is good, and the paper is easy to follow.

**Weaknesses:**

1. The novelty is limited. While execution-based evaluation is impactful, the conceptual novelty mainly lies in combining existing ideas (Chatbot Arena + executable sandbox). The methodology may be viewed more as engineering integration than a new algorithmic or theoretical contribution.
2. AUTOCODEARENA relies on LLM-as-a-Judge (Claude-3.7-Sonnet), which may itself introduce bias. The paper lacks calibration or agreement analysis between automated and human judgments.
3. Although expert volunteers are mentioned, the annotation process (e.g., inter-annotator agreement, quality metrics, error analysis) is not deeply evaluated, which could raise concerns about reliability.
4. I think the topic of the paper is not well-suited for ICLR. The paper looks more like a platform description rather than a research paper with clear methodological novelty.

**Questions:**

The author should provide more insight into the paper, for example, whether we can use the analysis behind the platform to further improve the current LLMs. Or how we can better help humans to use LLMs to generate code.

---

### Author Response · Authors · 2025-11-27
**Overall Response**

Dear Reviewers,

It is unfortunate that not all reviewers are in favor of such evaluation works, where the main contributions lie in creating a crowdsource platform like BigCodeArena and collecting data for experiments.

We thank every reviewer for the time reading our paper and sharing thoughts. We would like to clarify our contributions. First, we provide the first human preference crowdsourcing platforms for code generation that can execute a wide range of programming languages and frameworks, compared to other concurrent works like WebDevArena. Second, we collect a large-scale human preference dataset on code generation covering various languages, frameworks, and application scenarios, with detailed analysis in the Appendix. Third, we provide some practical usages of our datasets by constructing two benchmarks to evaluate existing LLMs.

From the perspective of Reviewer WBdo, we believe that there is not much we could address unless we start a completely new work. We note that even if works like ours are focused on engineering and evaluations but not training, they are still fit into ICLR, where there is a track called infrastructure, software libraries, hardware, etc. The reason that we submit it to the track datasets and benchmarks is the potential impact of the curated datasets we decide to release.

Regarding the vote quality, we have added the quality check results in Appendix G.4. We note that most details are documented in the Appendix, where we have stated in the main paper. For example, comparisons to existing leaderboards or ranking trends can be found in Appendix G.3.

---

### Note · Authors · 2026-01-02

I have read and agree with the venue's withdrawal policy on behalf of myself and my co-authors.